# TRAINABILITY PRESERVING NEURAL PRUNING

**Huan Wang**[1]  **Yun Fu**[1,2]
[1]Northeastern University, Boston, USA   [2]AInnovation Labs, Inc.
`wang.huan@northeastern.edu`   `yunfu@ece.neu.edu`

## ABSTRACT

Many recent works have shown *trainability* plays a central role in neural network pruning – unattended broken trainability can lead to severe under-performance and unintentionally amplify the effect of retraining learning rate, resulting in biased (or even misinterpreted) benchmark results. This paper introduces *trainability preserving pruning* (TPP), a scalable method to preserve network trainability against pruning, aiming for improved pruning performance and being more robust to retraining hyper-parameters (*e.g.*, learning rate). Specifically, we propose to penalize the *gram matrix* of convolutional filters to decorrelate the pruned filters from the retained filters. In addition to the convolutional layers, per the spirit of preserving the trainability of the whole network, we also propose to regularize the batch normalization parameters (scale and bias). Empirical studies on linear MLP networks show that TPP can perform on par with the oracle trainability recovery scheme. On nonlinear ConvNets (ResNet56/VGG19) on CIFAR10/100, TPP outperforms the other counterpart approaches by an obvious margin. Moreover, results on ImageNet-1K with ResNets suggest that TPP consistently performs more favorably against other top-performing structured pruning approaches. Code: https://github.com/MingSun-Tse/TPP.

## 1 INTRODUCTION

Neural pruning aims to remove redundant parameters without seriously compromising the performance. It normally consists of three steps (Reed, 1993; Han et al., 2015; 2016b; Li et al., 2017; Liu et al., 2019b; Wang et al., 2021b; Gale et al., 2019; Hoefler et al., 2021; Wang et al., 2023): *pretrain* a dense model; *prune* the unnecessary connections to obtain a sparse model; *retrain* the sparse model to regain performance. Pruning is usually categorized into two classes, unstructured pruning (*a.k.a.* element-wise pruning or fine-grained pruning) and structured pruning (*a.k.a.* filter pruning or coarse-grained pruning). Unstructured pruning chooses a single weight as the basic pruning element; while structured pruning chooses a group of weights (*e.g.*, 3d filter or a 2d channel) as the basic pruning element. Structured pruning fits more for acceleration because of the regular sparsity. Unstructured pruning, in contrast, results in irregular sparsity, hard to exploit for acceleration unless customized hardware and libraries are available (Han et al., 2016a; 2017; Wen et al., 2016).

Recent papers (Renda et al., 2020; Le & Hua, 2021) report an interesting phenomenon: During retraining, a larger learning rate (LR) helps achieve a *significantly better* final performance, empowering the two baseline methods, random pruning and magnitude pruning, to match or beat many more complex pruning algorithms. The reason behind is argued (Wang et al., 2021a; 2023) to be related to the *trainability* of neural networks (Saxe et al., 2014; Lee et al., 2020; Lubana & Dick, 2021). They make two major observations to explain the LR effect mystery (Wang et al., 2023). **(1)** The weight removal operation immediately breaks the network trainability or dynamical isometry (Saxe et al., 2014) (the ideal case of trainability) of the trained network. **(2)** The broken trainability slows down the optimization in retraining, where a greater LR aids the model converge faster, thus a better performance is observed earlier – using a smaller LR can actually do as well, but needs more epochs.

Although these works (Lee et al., 2020; Lubana & Dick, 2021; Wang et al., 2021a; 2023) provide a plausibly sound explanation, a more practical issue is *how to recover the broken trainability or maintain it during pruning*. In this regard, Wang et al. (2021a) proposes to apply *weight orthogonalization* based on QR decomposition (Trefethen & Bau III, 1997; Mezzadri, 2006) to the pruned

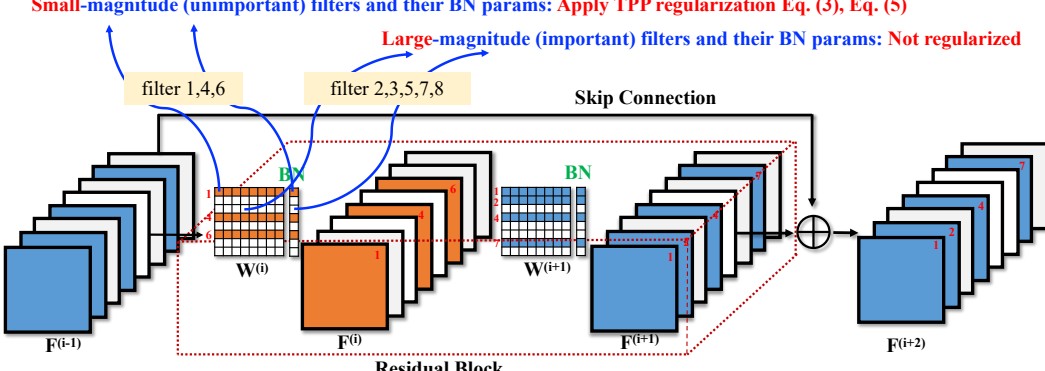

Figure 1: Illustration of the proposed TPP algorithm on a typical residual block. Weight parameters are classified into two groups as a typical pruning algorithm does: important (white color) and unimportant (orange or blue color), right from the beginning (before any training starts) based on the filter $L_1$-norms. Then *only* the unimportant parameters are enforced with the proposed TPP regularization terms, which is the key to maintain trainability when the unimportant weights are eventually eliminated from the network. Notably, the critical part of a regularization-based pruning algorithm lies in its *specific regularization term*, *i.e.*, Eqs. (3) and (5), which we will show perform more favorably than other alternatives (see Tabs. 1 and 10).

model. However, their method is shown to only work for *linear* MLP networks. On modern deep convolutional neural networks (CNNs), how to maintain trainability during pruning is still elusive.

We introduce *trainability preserving pruning* (TPP), a new and novel filter pruning algorithm (see Fig. 1) that maintains trainability via a regularized training process. By our observation, the primary cause that pruning breaks trainability lies in the dependency among parameters. The primary idea of our approach is thus to *decorrelate* the pruned weights from the kept weights so as to "cut off" the dependency, so that the subsequent sparsifying operation barely hurts the network trainability.

Specifically, we propose to regularize the *gram matrix* of weights: All the entries representing the correlation between the pruned filters (*i.e.*, unimportant filters) and the kept filters (*i.e.*, important filters) are encouraged to diminish to zero. This is the first technical contribution of our method. The second one lies in how to treat the other entries. Conventional dynamical isometry wisdom suggests *orthogonality*, namely, 1 self-correlation and 0 cross-correlation, even among the kept filters, while we find directly translating the orthogonality idea here is *unnecessary or even harmful* because the too strong penalty will constrain the optimization, leading to deteriorated local minimum. Rather, we propose *not to impose any regularization* on the correlation entries of kept filters.

Finally, modern deep models are typically equipped with batch normalization (BN) (Ioffe & Szegedy, 2015). However, previous filter pruning papers rarely explicitly take BN into account (except two (Liu et al., 2017; Ye et al., 2018); the differences of our work from theirs will be discussed in Sec. 3.2) to mitigate the side effect when it is removed because its associated filter is removed. Since they are also a part of the whole trainable parameters in the network, unattended removal of them will also lead to severely crippled trainability (especially at large sparsity). Therefore, BN parameters (both the scale and bias included) ought to be explicitly taken into account too, when we develop the pruning algorithm. Based on this idea, we propose to regularize the two learnable parameters of BN to minimize the influence of its absence later.

Practically, our TPP is easy to implement and robust to hyper-parameter variations. On ResNet50 ImageNet, TPP delivers encouraging results compared to many recent SOTA filter pruning methods.

**Contributions.** **(1)** We present the *first* filter pruning method (*trainability preserving pruning*) that effectively maintains trainability during pruning for modern deep networks, via a customized weight gram matrix as regularization target. **(2)** Apart from weight regularization, a BN regularizer is introduced to allow for their subsequent absence in pruning – this issue has been overlooked by most previous pruning papers, although it is shown to be pretty important to preserve trainability, especially in the large sparsity regime. **(3)** Practically, the proposed method can easily scale to

modern deep networks (such as ResNets) and datasets (such as ImageNet-1K (Deng et al., 2009)). It achieves promising pruning performance in the comparison to many SOTA filter pruning methods.

## 2 RELATED WORK

**Network pruning**. Pruning mainly falls into structured pruning (Li et al., 2017; Wen et al., 2016; He et al., 2017; 2018; Wang et al., 2021b) and unstructured pruning (Han et al., 2015; 2016b; LeCun et al., 1990; Hassibi & Stork, 1993; Singh & Alistarh, 2020), according to the sparsity structure. For more comprehensive coverage, we recommend surveys (Sze et al., 2017; Cheng et al., 2018; Deng et al., 2020; Hoefler et al., 2021; Wang et al., 2022). This paper targets structured pruning (*filter pruning*, to be specific) because it is more imperative to make modern networks (*e.g.*, ResNets (He et al., 2016)) *faster* rather than *smaller* compared to the early single-branch convolutional networks.

It is noted that random pruning of a normally-sized (*i.e.*, not severely over-parameterized) network usually leads to significant performance drop. We need to cleverly choose some unimportant parameters to remove. Such a criterion for choosing is called *pruning criterion*. In the area, there have been two major paradigms to address the pruning criterion problem dating back to the 1990s: *regularization-based* methods and *importance-based* (*a.k.a.* saliency-based) methods (Reed, 1993).

Specifically, the regularization-based approaches choose unimportant parameters via a sparsity-inducing penalty term (*e.g.*, Wen et al. (2016); Yang et al. (2020); Lebedev & Lempitsky (2016); Louizos et al. (2018); Liu et al. (2017); Ye et al. (2018); Zhang et al. (2021a; 2022; 2021b)). This paradigm can be applied to a random or pretrained network. Importance-based methods choose unimportant parameters via an importance formula, derived from the Taylor expansion of the loss function (*e.g.*, LeCun et al. (1990); Hassibi & Stork (1993); Han et al. (2015; 2016b); Li et al. (2017); Molchanov et al. (2017; 2019)). This paradigm is majorly applied to a pretrained network. Despite the differences, it is worth noting that these two paradigms are *not* firmly unbridgeable. We can develop approaches that take advantage of *both* ideas, such as Ding et al. (2018); Wang et al. (2019; 2021b) – these methods identify unimportant weights per a certain importance criterion; then, they utilize a penalty term to produce sparsity. Our TPP method in this paper is also in this line.

**Trainability, dynamical isometry, and orthogonality**. Trainability describes the easiness of optimization of a neural network. Dynamical isometry, the perfect case of trainability, is first introduced by Saxe et al. (2014), stating that singular values of the Jacobian matrix are close to 1. It can be achieved (for linear MLP models) by the orthogonality of weight matrix at initialization. Recent works on this topic mainly focus on how to maintain dynamical isometry *during training* instead of only for initialization (Xie et al., 2017; Huang et al., 2018; Bansal et al., 2018; Huang et al., 2020; Wang et al., 2020). These methods are developed independent of pruning, thus not directly related to our proposed approach. However, the insights from these works inspire us to our method (see Sec. 3.2) and possibly more in the future. Several pruning papers study the network trainability issue in the context of network pruning, such as Lee et al. (2020); Lubana & Dick (2021); Vysogorets & Kempe (2021). These works mainly discuss the trainability issue of a *randomly* initialization network. In contrast, we focus on the pruning case of a *pretrained* network.

## 3 METHODOLOGY

### 3.1 PREREQUISITES: DYNAMICAL ISOMETRY AND ORTHOGONALITY

The definition of dynamical isometry is that the Jacobian of a network has as many singular values (JSVs) as possible close to 1 (Saxe et al., 2014). With it, the error signal can preserve its norm during propagation without serious amplification or attenuation, which in turn helps the convergence of (very deep) networks. For a single fully-connected layer $W$, a sufficient and necessary condition to realize dynamical isometry is orthogonality, *i.e.*, $W^\top W = I$,

$$
\begin{aligned}
\mathbf{y} &= W\mathbf{x}, \\
||\mathbf{y}|| &= \sqrt{\mathbf{y}^\top \mathbf{y}} = \sqrt{\mathbf{x}^\top W^\top W \mathbf{x}} = ||\mathbf{x}||, \ \ \textit{iff.} \ W^\top W = I,
\end{aligned}
\tag{1}
$$

where $I$ represents the *identity* matrix. Orthogonality of a weight matrix can be easily realized by matrix orthogonalization techniques such as QR decomposition (Trefethen & Bau III, 1997; Mezzadri, 2006). *Exact* (namely all the Jacobian singular values are exactly 1) dynamical isometry

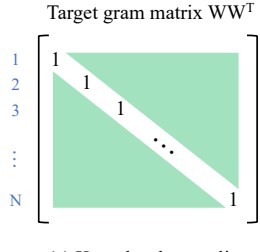
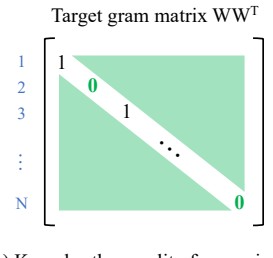
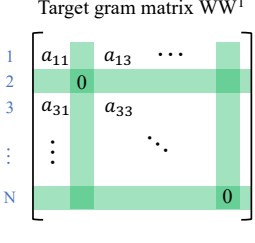

(a) Kernel orthogonality      (b) Kernel orthogonality for pruning      (c) De-correlate pruned from kept

Figure 2: Regularization target comparison between the proposed scheme (c) and similar counterparts (a) and (b). Green part stands for zero entries. Index 1 to N denotes the filter indices. In (b, c), filter 2 and N are the unimportant filters to be removed. **(a)** Regularization target of pure kernel orthogonality (an identity matrix), no pruning considered. **(b)** Regularization target of directly applying the weight orthogonality to filter pruning. **(c)** Regularization target of the proposed *weight de-correlation* solution in TPP: only regularize the filters to be removed, leave the others unconstrained. This scheme maintains trainability while imposing the least constraint on the weights.

can be achieved for *linear* networks since multiple linear layers essentially reduce to a single 2d weight matrix. In contrast, the convolutional and non-linear cases are much complicated. Previous work (Wang et al., 2023) has shown that merely considering convolution or ReLU (Nair & Hinton, 2010) renders the weight orthogonalization method much less effective in terms of recovering dynamical isometry after pruning, let alone considering modern deep networks with BN (Ioffe & Szegedy, 2015) and residuals (He et al., 2016). The primary goal of our paper is to bridge this gap.

Following the seminal work of Saxe et al. (2014), several papers propose to maintain orthogonality *during training* instead of sorely for the initialization. There are primarily two groups of orthogonalization methods for CNNs: kernel orthogonality (Xie et al., 2017; Huang et al., 2018; 2020) and orthogonal convolution (Wang et al., 2020),

$$
\begin{aligned}
KK^\top = I &\Rightarrow \mathcal{L}_{orth} = KK^\top - I, \quad \triangleleft \textbf{ kernel orthogonality} \\
\mathcal{K}\mathcal{K}^\top = I &\Rightarrow \mathcal{L}_{orth} = \mathcal{K}\mathcal{K}^\top - I. \quad \triangleleft \textbf{ orthogonal convolution}
\end{aligned}
\tag{2}
$$

Clearly the difference lies in the weight matrix $K$ *vs.* $\mathcal{K}$: **(1)** $K$ denotes the original weight matrix in a convolutional layer. Weights of a CONV layer make up a 4d tensor $\mathbb{R}^{N \times C \times H \times W}$ ($N$ stands for the output channel number, $C$ for the input channel number, $H$ and $W$ for the height and width of the CONV kernel). Then, $K$ is a reshaped version of the 4d tensor: $K \in \mathbb{R}^{N \times CHW}$ (if $N < CHW$; otherwise, $K \in \mathbb{R}^{CHW \times N}$). **(2)** In contrast, $\mathcal{K} \in \mathbb{R}^{NH_{fo}W_{fo} \times CH_{fi}W_{fi}}$ stands for the *doubly block-Toeplitz* representation of $K$ ($H_{fo}$ stands for the *output* feature map height, $H_{fi}$ for the *input* feature map height. $W_{fo}$ and $W_{fi}$ can be inferred the same way for width).

Wang et al. (2020) have shown that orthogonal convolution is more effective than kernel orthogonality (Xie et al., 2017) in that the latter is only a necessary but insufficient condition of the former. In this work, we will evaluate *both* methods to see how effective they are in recovering trainability.

### 3.2 TRAINABILITY PRESERVING PRUNING (TPP)

Our TPP method has two parts. First, we explain how we come up with the proposed scheme and how it intuitively is better than the straight idea of directly applying orthogonality regularization methods (Xie et al., 2017; Wang et al., 2020) here. Second, a batch normalization regularizer is introduced given the prevailing use of BN as a standard component in deep CNNs nowadays.

**(1) Trainability *vs.* orthogonality**. From previous works (Lee et al., 2020; Lubana & Dick, 2021; Wang et al., 2021a; 2023), we know recovering the broken trainability (or dynamical isometry) impaired by pruning is very important. Considering orthogonality regularization can encourage isometry, a pretty straightforward solution is to build upon the existing weight orthogonality regularization schemes. Specifically, kernel orthogonality regularizes the weight gram matrix towards an *identity matrix* (see Fig. 2(a)). In our case, we aim to remove some filters, so naturally we can regularize the weight gram matrix to be close to a *partial identity matrix*, with the diagonal entries at the pruned filters zeroed (see Fig. 2(b); note the diagonal green zeros).

The above scheme is simple and straightforward. However, it is not in the best shape by our empirical observation. It imposes too strong *unnecessary* constraint on the remaining weights, which will in turn hurt the optimization. Therefore, we propose to seek a *weaker* constraint, not demanding the perfect trainability (*i.e.*, exact isometry realized by orthogonality), but only a *benign* status, which describes a state of the neural network where gradients can flow effectively through the model without being interrupted. Orthogonality requires the Jacobian singular values to be exactly 1; in contrast, a benign trainability only requires them *not to be extremely large or small* so that the network can be trained normally. To this end, we propose to *decorrelate* the kept filters from the pruned ones: in the target gram matrix, all the entries associated with the pruned filters are zero; all the other entries stay *as they are* (see Fig. 2(c)). This scheme will be empirically justified (Tab. 3).

Specifically, all the filters in a layer are sorted based on their $L_1$-norms. Then, we consider those with the smallest $L_1$ norms as *unimportant filters* (the $S_l$ below) (so the proposed method also falls into the magnitude-based pruning method group). Then, the proposed regularization term is,

$$\mathcal{L}_1 = \sum_{l=1}^{L} ||W_l W_l^\top \odot (\mathbf{1} - \mathbf{m}\mathbf{m}^\top)||_F^2, \ \mathbf{m}_j = 0 \text{ if } j \in S_l, \text{ else } 1, \tag{3}$$

where $W$ refers to the weight matrix; $\mathbf{1}$ represents the matrix full of 1; $\mathbf{m}$ is a 0/1-valued column mask vector; $\odot$ is the Hadamard (element-wise) product; and $|| \cdot ||_F$ denotes the Frobenius norm.

**(2) BN regularization**. Per the idea of preserving trainability, BN is not ignorable since BN layers are also trainable. Removing filters will change the internal feature distributions. If the learned BN statistics do not change accordingly, the error will accumulate and result in deteriorated performance (especially for deep networks). Consider the following BN formulation (Ioffe & Szegedy, 2015),

$$f = \gamma \frac{W * X - \mu}{\sqrt{\sigma^2 + \epsilon}} + \beta, \tag{4}$$

where $*$ stands for convolution; $\mu/\sigma^2$ refers to the running mean/variance; $\epsilon$, a small number, is used for numerical stability. The two learnable parameters are $\gamma$ and $\beta$. Although unimportant weights are enforced with regularization for sparsity, their magnitude can barely be exact zero, making the subsequent removal of filters *biased*. This will skew the feature distribution and render the BN statistics inaccurate. Using these biased BN statistics will be improper and damages trainability. To mitigate such influence from BN, we propose to regularize *both* the $\gamma$ and $\beta$ of *pruned* feature map channels to zero, which gives us the following BN penalty term,

$$\mathcal{L}_2 = \sum_{l=1}^{L} \sum_{j \in S_l} \gamma_j^2 + \beta_j^2. \tag{5}$$

The merits of BN regularization will be justified in our experiments (Tab. 4).

To sum, with the proposed regularization terms, the total error function is

$$\mathcal{E} = \mathcal{L}_{cls} + \frac{\lambda}{2}(\mathcal{L}_1 + \mathcal{L}_2), \tag{6}$$

where $\mathcal{L}_{cls}$ means the original classification loss. The coefficient $\lambda$ *grows* by a predefined constant $\Delta$ per $K_u$ iterations (up to a ceiling $\tau$) during training to ensure the pruned parameters are rather close to zero (inspired by Wang et al. (2019; 2021b)). See Algorithm 1 in Appendix for more details.

**Discussion**. Prior works (Liu et al., 2017; Ye et al., 2018) also study regularizing BN for pruning. Our BN regularization method is *starkly different* from theirs. **(1)** In terms of the motivation or goal, Liu et al. (2017); Ye et al. (2018) regularize $\gamma$ to *learn* unimportant filters, namely, regularizing BN is to indirectly decide which filters are unimportant. In contrast, in our method, unimportant filters are decided by their $L_1$-norms. We adopt BN regularization for a totally different consideration – to mitigate the side effect of breaking trainability, which is not mentioned at all in their works. **(2)** In terms of specific technique, Liu et al. (2017); Ye et al. (2018) only regularize the multiplier factor $\gamma$ (because it is enough to decide which filters are unimportant) while we regularize *both* the learnable parameters because only regularizing one still misses a few trainable parameters. Besides, we employ different regularization strength for different parameters (by the group of important filters *vs.* unimportant filters), while Liu et al. (2017); Ye et al. (2018) simply adopt a uniform penalty

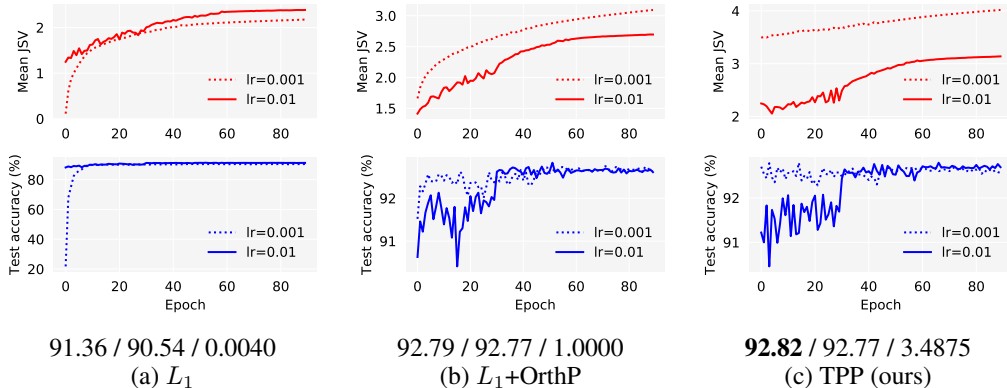

| 91.36 / 90.54 / 0.0040 | 92.79 / 92.77 / 1.0000 | **92.82** / 92.77 / 3.4875 |
| (a) $L_1$ | (b) $L_1$+OrthP | (c) TPP (ours) |

Figure 3: Mean Jacobian singular value (JSV) and test accuracy during retraining with different setups (network: MLP-7-Linear, dataset: MNIST). Below each plot are, in order, the best accuracy of LR `1e-2`, the best accuracy of LR `1e-3`, and the mean JSV right after pruning (*i.e.*, without retraining). LR `1e-2` and `1e-3` are short for two retraining LR schedules: {init LR 1e-2, decay at epoch 30/60, epochs:90}, {init LR 1e-3, decay at epoch 45, epochs:90}. The accuracies are averaged by 5 random runs. For reference, the unpruned model has mean JSV 2.4987, test accuracy 92.77.

strength for *all* parameters – this is another key difference because regularizing all parameters (including those that are meant to be retained) will damage trainability, which is exactly what we want to avoid. In short, in terms of either general motivation or specific technical details, our proposed BN regularization is *distinct* from previous works (Liu et al., 2017; Ye et al., 2018).

## 4 EXPERIMENTS

**Networks and datasets**. We first present some analyses with MLP-7-Linear network on MNIST (LeCun et al., 1998). Then compare our method to other plausible solutions with the ResNet56 (He et al., 2016) and VGG19 (Simonyan & Zisserman, 2015) networks, on the CIFAR10 and 100 datasets (Krizhevsky, 2009), respectively. Next we evaluate our algorithm on ImageNet-1K (Deng et al., 2009) with ResNet34 and ResNet50 (He et al., 2016). Finally, we present ablation studies to show the efficacy of two main technical novelties in our approach. On ImageNet, we use public *torchvision* models (Paszke et al., 2019) as the unpruned models for fair comparison with other papers. On other datasets, we train our own base models with comparable accuracies reported in their original papers. See the Appendix (Tab. 5) for concrete training settings.

**Comparison methods**. We compare with Wang et al. (2021a), which proposes a method, *OrthP*, to recover broken trainability after pruning *pretrained* models. Furthermore, since weight orthogonality is closely related to network trainability and there have been plenty of orthogonality regularization approaches (Xie et al., 2017; Wang et al., 2020; Huang et al., 2018; 2020; Wang et al., 2020), a straightforward solution is to combine them with $L_1$ pruning (Li et al., 2017) to see whether they can help maintain or recover the broken trainability. Two plausible combination schemes are easy to see: **1)** apply orthogonality regularization methods *before* $L_1$ pruning, **2)** apply orthogonality regularization methods *after* $L_1$ pruning, *i.e.*, in retraining. Two representative orthogonality regularization methods are selected because of their proved effectiveness: kernel orthogonality (KernOrth) (Xie et al., 2017) and convolutional orthogonality (OrthConv) (Wang et al., 2020), so in total there are four combinations: $L_1$ + KernOrth, $L_1$ + OrthConv, KernOrth + $L_1$, OrthConv + $L_1$.

**Comparison metrics**. (1) We examine the final test accuracy *after retraining* with the similar FLOPs budget – this is currently the most prevailing metric to compare different filter pruning methods in classification. Concretely, we compare two settings: a relatively large retraining LR (`1e-2`) and a small one (`1e-3`). We introduce these settings because previous works (Renda et al., 2020; Le & Hua, 2021; Wang et al., 2021a; 2023) have showed that retraining LR has a great impact on the final performance. From this metric, we can see how sensitive different methods are to the retraining LR. (2) We also compare the test accuracy *before retraining* – from this metric, we will see how robust different methods are in the face of weight removal.

Table 1: Test accuracy (%) comparison among different isometry maintenance or recovery methods on ResNet56 on CIFAR10. *Scratch* stands for training from scratch. *KernOrth* means Kernel Orthogonalization (Xie et al., 2017); *OrthConv* means Convolutional Orthogonalization (Wang et al., 2020). Two retraining LR schedules are evaluated here: initial LR `1e-2` *vs.* `1e-3`. *Acc. diff.* refers to the accuracy gap of LR `1e-3` against LR `1e-2`.

| ResNet56 on CIFAR10: Unpruned acc. 93.78%, Params: 0.85M, FLOPs: 0.25G | | | | | |
|---|---|---|---|---|---|
| Layerwise PR | 0.3 | 0.5 | 0.7 | 0.9 | 0.95 |
| Sparsity/Speedup | 31.14%/1.45× | 49.82%/1.99× | 70.57%/3.59× | 90.39%/11.41× | 95.19%/19.31× |
| | | Initial | retraining | LR 1e-2 | |
| Scratch | 93.16 (0.16) | 92.78 (0.23) | 92.11 (0.12) | 88.36 (0.20) | 84.60 (0.14) |
| $L_1$ (Li et al., 2017) | 93.79 (0.06) | **93.51** (0.07) | 92.26 (0.17) | 86.75 (0.31) | 83.03 (0.07) |
| $L_1$ + OrthP (Wang et al., 2021a) | 93.69 (0.02) | 93.36 (0.19) | 91.96 (0.06) | 86.01 (0.34) | 82.62 (0.05) |
| $L_1$ + KernOrth (Xie et al., 2017) | 93.49 (0.04) | 93.30 (0.19) | 91.71 (0.14) | 84.78 (0.34) | 80.87 (0.47) |
| $L_1$ + OrthConv (Wang et al., 2020) | 92.54 (0.09) | 92.41 (0.07) | 91.02 (0.16) | 84.52 ( 0.27) | 80.23 (1.19) |
| KernOrth (Xie et al., 2017) + $L_1$ | 93.49 (0.07) | 92.82 (0.10) | 90.54 (0.25) | 85.47 (0.20) | 79.48 (0.81) |
| OrthConv (Wang et al., 2020) + $L_1$ | 93.63 (0.17) | 93.28 (0.20) | 92.27 (0.13) | 86.70 (0.07) | 83.21 (0.61) |
| **TPP** (ours) | **93.81** (0.11) | 93.46 (0.06) | **92.35** (0.12) | **89.63** (0.10) | **85.86** (0.08) |
| | | Initial | retraining | LR 1e-3 | |
| $L_1$ (Li et al., 2017) | 93.43 (0.06) | 93.12 (0.10) | 91.77 (0.11) | 87.57 (0.09) | 83.10 (0.12) |
| **TPP** (ours) | **93.54** (0.08) | **93.32** (0.11) | **92.00** (0.08) | **89.09** (0.10) | **85.47** (0.22) |
| Acc. diff. ($L_1$) | -0.38 | -0.40 | -0.50 | +0.82 | +0.07 |
| Acc. diff. (TPP) | -0.27 | -0.14 | -0.35 | -0.54 | -0.39 |

## 4.1 ANALYSIS: MLP-7-LINEAR ON MNIST AND RESNET56 ON CIFAR10

*MLP-7-Linear* is a seven-layer linear MLP. It is adopted in Wang et al. (2021a) for analysis because linear MLP is the only network that can achieve *exact* dynamical isometry (all JSVs are exactly 1) so far. Their proposed dynamical isometry recovery method, OrthP (Wang et al., 2021a), is shown to achieve exact isometry on linear MLP networks. Since we claim our method TPP can maintain dynamical isometry too, conceivably, our method should play a similar role to OrthP in pruning. To confirm this, we prune the MLP-7-Linear network with our method.

**TPP can perform as well as OrthP on linear MLP**. In Fig. 3, (b) is the one equipped with OrthP, which can *exactly* recover dynamical isometry (note its mean JSV right after pruning is 1.0000), so it works as the oracle here. **(1)** OrthP improves the best accuracy from 91.36/90.54 to 92.79/92.77. Using TPP, we obtain 92.81/92.77. Namely, in terms of accuracy, our method is as good as the oracle scheme. **(2)** Note the mean JSV right after pruning – the $L_1$ pruning destroys the mean JSV from 2.4987 to 0.0040, and OrthP brings it back to 1.0000. In comparison, TPP achieves 3.4875, at the same order of magnitude of 1.0000, also as good as OrthP. These demonstrate, in terms of either the final evaluation metric (test accuracy) or the trainability measure (mean JSV), our TPP performs as well as the oracle method OrthP on the linear MLP.

**Loss surface analysis with ResNet56 on CIFAR10**. We further analyze the loss surfaces (Li et al., 2018) of pruned networks (before retraining) by different methods. Our result (due to limited space, we defer this result to Appendix; see Fig. 4) suggests that the loss surface of our method is *flatter* than other methods, implying the loss landscape is *easier* for optimization.

## 4.2 RESNET56 ON CIFAR10 / VGG19 ON CIFAR100

Here we compare our method to other plausible solutions on the CIFAR datasets (Krizhevsky, 2009) with non-linear convolutional architectures. The results in Tab. 1 (for CIFAR10) and Tab. 10 (for CIFAR100, deferred to Appendix due to limited space here) show that,

(1) OrthP does not work well – $L_1$ + OrthP underperforms the original $L_1$ under all the five pruning ratios for both ResNet56 and VGG19. This further confirms the weight orthogonalization method proposed for linear networks indeed does not generalize to non-linear CNNs.

(2) For KernOrth *vs.* OrthConv, the results look mixed – OrthConv is generally better when applied *before* the $L_1$ pruning. This is reasonable since OrthConv is shown more effective than KernOrth in enforcing more isometry (Wang et al., 2020), which in turn can stand more damage of pruning.

(3) Of particular note is that, none of the above five methods actually outperform the $L_1$ pruning or the simple scratch training. It means that neither enforcing more isometry before pruning nor

Table 2: Comparison on ImageNet-1K validation set. *Advanced training recipe (such as cosine LR schedule) is used; we single them out for fair comparison.

| Method | Model | Unpruned top-1 (%) | Pruned top-1 (%) | Top-1 drop (%) | Speedup |
|---|---|---|---|---|---|
| $L_1$ (pruned-B) (Li et al., 2017) | | 73.23 | 72.17 | 1.06 | **1.32×** |
| $L_1$ (pruned-B, reimpl.) (Wang et al., 2023) | | 73.31 | 73.67 | -0.36 | **1.32×** |
| Taylor-FO (Molchanov et al., 2019) | ResNet34 | 73.31 | 72.83 | 0.48 | 1.29× |
| GReg-2 (Wang et al., 2021b) | | 73.31 | 73.61 | -0.30 | **1.32×** |
| **TPP** (ours) | | 73.31 | **73.77** | **-0.46** | **1.32×** |
| ProvableFP (Liebenwein et al., 2020) | | 76.13 | 75.21 | 0.92 | 1.43× |
| MetaPruning (Liu et al., 2019a) | ResNet50 | 76.6 | 76.2 | 0.4 | 1.37× |
| GReg-1 (Wang et al., 2021b) | | 76.13 | 76.27 | -0.14 | **1.49×** |
| **TPP** (ours) | | 76.13 | **76.44** | **-0.31** | **1.49×** |
| IncReg (Wang et al., 2019) | | 75.60 | 72.47 | 3.13 | 2.00× |
| SFP (He et al., 2018) | | 76.15 | 74.61 | 1.54 | 1.72× |
| HRank (Lin et al., 2020) | | 76.15 | 74.98 | 1.17 | 1.78× |
| Taylor-FO (Molchanov et al., 2019) | | 76.18 | 74.50 | 1.68 | 1.82× |
| Factorized (Li et al., 2019) | | 76.15 | 74.55 | 1.60 | **2.33×** |
| DCP (Zhuang et al., 2018) | ResNet50 | 76.01 | 74.95 | 1.06 | 2.25× |
| CCP-AC (Peng et al., 2019) | | 76.15 | 75.32 | 0.83 | 2.18× |
| GReg-2 (Wang et al., 2021b) | | 76.13 | 75.36 | 0.77 | 2.31× |
| CC (Li et al., 2021) | | 76.15 | 75.59 | 0.56 | 2.12× |
| MetaPruning (Liu et al., 2019a) | | 76.6 | 75.4 | 1.2 | 2.00× |
| **TPP** (ours) | | 76.13 | **75.60** | **0.53** | 2.31× |
| LFPC (He et al., 2020) | | 76.15 | 74.46 | 1.69 | 2.55× |
| GReg-2 (Wang et al., 2021b) | ResNet50 | 76.13 | 74.93 | 1.20 | 2.56× |
| CC (Li et al., 2021) | | 76.15 | 74.54 | 1.61 | **2.68×** |
| **TPP** (ours) | | 76.13 | **75.12** | **1.01** | 2.56× |
| IncReg (Wang et al., 2019) | | 75.60 | 71.07 | 4.53 | 3.00× |
| Taylor-FO (Molchanov et al., 2019) | ResNet50 | 76.18 | 71.69 | 4.49 | 3.05× |
| GReg-2 (Wang et al., 2021b) | | 76.13 | 73.90 | 2.23 | **3.06×** |
| **TPP** (ours) | | 76.13 | **74.51** | **1.62** | **3.06×** |

| Method | Network | Top-1 (%) | FLOPs (G) |
|---|---|---|---|
| CHEX* (Hou et al., 2022) | | 77.4 | 2 |
| CHEX* (Hou et al., 2022) | ResNet50 | 76.0 | 1 |
| **TPP*** (ours) | | **77.75** | 2 |
| **TPP*** (ours) | | **76.52** | 1 |

compensating isometry after pruning can help recover trainability. In stark contrast, our TPP method outperforms $L_1$ pruning and scratch *consistently against different pruning ratios* (only one exception is pruning ratio 0.7 on ResNet56, but our method is still the second best and the gap to the best is only marginal: 93.46 *vs.* 93.51). Besides, note that the accuracy trend – in general, with a *larger* sparsity ratio, TPP beats $L_1$ or Scratch by a *more pronounced* margin. This is because, ar a larger pruning ratio, the trainability is impaired more, where our method can help more, thus harvesting more performance gains. We will see similar trends many times.

(4) In Tabs. 1 and 10, we also present the results when the initial retraining LR is `1e-3`. Wang et al. (2021a) argue that if the broken dynamical isometry can be well maintained/recovered, the final performance gap between LR `1e-2` and `1e-3` should be diminished. Now that TPP is claimed to be able to maintain trainability, the performance gap should become smaller. This is empirically verified in the table. In general, the accuracy gap between LR `1e-2` and LR `1e-3` of TPP is smaller than that of $L_1$ pruning. Two exceptions are PR 0.9/0.95 on ResNet56: LR `1e-3` is unusually better than LR `1e-2` for $L_1$ pruning. Despite them, the general picture is that the accuracy gap between LR `1e-3` and `1e-2` turns smaller with TPP. This is a sign that trainability is effectively maintained.

### 4.3 IMAGENET BENCHMARK

We further evaluate TPP on ImageNet-1K (Deng et al., 2009) in comparison to many existing filter pruning algorithms. Results in Tab. 2 show that TPP is *consistently* better than the others across different speedup ratios. Moreover, under larger speedups, the advantage of our method is usually more evident. *E.g.*, TPP outperforms Taylor-FO (Molchanov et al., 2019) by 1.15% in terms of the top-1 acc. drop at the 2.31×speedup track; at 3.06×speedup, TPP leads Taylor-FO (Molchanov et al., 2019) by 2.87%. This shows *TPP is more robust to more aggressive pruning*. The reason is easy to see – more aggressive pruning hurts trainability more (Lee et al., 2019; Wang et al., 2023), where our method can find more use, in line with the observations on CIFAR (Tabs. 1 and 10).

Table 3: Test accuracy (without retraining) comparison between two plausible schemes *diagonal vs. decorrelate* in our TPP method.

| ResNet56 on CIFAR10: Unpruned acc. 93.78%, Params: 0.85M, FLOPs: 0.25G | | | | | |
|---|---|---|---|---|---|
| Layerwise PR | 0.3 | 0.5 | 0.7 | 0.9 | 0.95 |
| TPP (diagonal) | 92.67 (0.29) | 91.97 (0.02) | **90.21** (0.23) | 23.23 (5.19) | 14.23 (1.42) |
| TPP (decorrelate) | **92.74** (0.16) | **92.07** (0.05) | 89.95 (0.26) | **30.35** (4.69) | **17.33** (0.50) |
| VGG19 on CIFAR100: Unpruned acc. 74.02%, Params: 20.08M, FLOPs: 0.80G | | | | | |
| Layerwise PR | 0.1 | 0.3 | 0.5 | 0.7 | 0.9 |
| TPP (diagonal) | 68.70 (0.18) | 64.55 (0.14) | 55.66 (0.73) | 13.76 (0.53) | 1.00 (0.00) |
| TPP (decorrelate) | **72.43** (0.12) | **69.31** (0.11) | **62.59** (0.14) | **18.97** (1.25) | 1.00 (0.00) |

Table 4: Test accuracy (without retraining) comparison *w.r.t.* the proposed weight gram matrix regularization and BN regularization. PR stands for layerwise pruning ratio.

| ResNet56 on CIFAR10: Unpruned acc. 93.78%, Params: 0.85M, FLOPs: 0.25G | | | | | | |
|---|---|---|---|---|---|---|
| Gram Reg | BN Reg | PR = 0.3 | PR = 0.5 | PR = 0.7 | PR = 0.9 | PR = 0.95 |
| ✓ | ✓ | **92.94** (0.14) | **92.48** (0.19) | **90.48** (0.09) | **70.53** (1.69) | **23.05** (2.61) |
| ✓ | ✗ | 92.79 (0.03) | 92.23 (0.08) | 90.46 (0.21) | 44.25 (2.46) | 16.52 (0.43) |
| ✗ | ✓ | 92.40 (0.30) | 91.95 (0.04) | 90.26 (0.23) | 26.79 (2.19) | 10.50 (0.63) |
| VGG19 on CIFAR100: Unpruned acc. 74.02%, Params: 20.08M, FLOPs: 0.80G | | | | | | |
| Gram Reg | BN Reg | PR = 0.1 | PR = 0.3 | PR = 0.5 | PR = 0.7 | PR = 0.9 |
| ✓ | ✓ | **73.44** (0.07) | **71.61** (0.12) | **69.28** (0.25) | **65.15** (0.20) | **2.84** (1.13) |
| ✓ | ✗ | 73.01 (0.13) | 71.26 (0.19) | 68.67 (0.10) | 61.70 (0.46) | 1.75 (0.38) |
| ✗ | ✓ | 71.97 (0.23) | 70.26 (0.61) | 68.40 (0.30) | 2.10 (0.27) | 1.02 (0.03) |

We further compare to more strong pruning methods. Notably, DMCP (Guo et al., 2020), LeGR (Chin et al., 2020), EagleEye (Li et al., 2020), and CafeNet (Su et al., 2021) have been shown *outperformed* by CHEX (Hou et al., 2022) (see their Tab. 1) with ResNet50 on ImageNet. Therefore, here we only compare to CHEX. Following CHEX, we employ more advanced training recipe (*e.g.*, cosine LR schedule) referring to TIMM (Wightman et al., 2021). Results in Tab. 2 suggest that our method *surpasses* CHEX at different FLOPs.

### 4.4 ABLATION STUDY

This section presents ablation studies to demonstrate the merits of TPP's two major innovations: **(1)** We propose not to over-penalize the kept weights in orthogonalization (*i.e.*, (c) *vs.* (b) in Fig. 2). **(2)** We propose to regularize the two learnable parameters in BN.

The results are presented in Tabs. 3 and 4, where we compare the accuracy right after pruning (*i.e.*, without retraining). We have the following major observations: **(1)** Tab. 3 shows using *decorrelate* (Fig. 2(c)) is better than using *diagonal* (Fig. 2(b)), generally speaking. Akin to Tabs. 1 and 10, at a greater sparsity ratio, the advantage of *decorrelate* is more pronounced, except for too large sparsity (0.95 for ResNet56, 0.9 for VGG19) because too large sparsity will break the trainability beyond repair. **(2)** For BN regularization, in Tab. 4, when it is switched off, the performance degrades. It also poses the similar trend: BN regularization is *more helpful* under the *larger sparsity*.

### 5 CONCLUSION

Trainability preserving is shown to be critical in neural network pruning, while few works have realized it on the modern large-scale non-linear deep networks. Towards this end, we present a new filter and novel pruning method named *trainability preserving pruning* (TPP) based on regularization. Specifically, we propose an improved weight gram matrix as regularization target, which does not unnecessarily over-penalize the retained important weights. Besides, we propose to regularize the BN parameters to mitigate its damage to trainability. Empirically, TPP performs as effectively as the ground-truth trainability recovery method and is more effective than other counterpart approaches based on weight orthogonality. Furthermore, on the standard ImageNet-1K benchmark, TPP also matches or even beats many recent SOTA filter pruning approaches. **As far as we are concerned, TPP is the *first* approach that explicitly tackles the *trainability preserving* problem in structured pruning that easily scales to the large-scale datasets and networks.**

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

## A    IMPLEMENTATION DETAILS

**Code reference**. We mainly refer to the following code implementations in this work. They are all open-licensed.

- Official PyTorch ImageNet example[1];
- GReg-1/GReg-2 (Wang et al., 2021b)[2];
- OrthConv (Wang et al., 2020)[3];
- Rethinking the value of network pruning (Liu et al., 2019b)[4].

**Data split**. All the datasets in this paper are public datasets with standard APIs in PyTorch (Paszke et al., 2019). We employs these standard APIs for the train/test data split to keep fair comparison with other methods.

**Training setups and hyper-parameters**. Tab. 5 summarizes the detailed training setups. For the hyper-parameters that are introduced in our TPP method: regularization granularity $\Delta$, regularization ceiling $\tau$ and regularization update interval $K_u$, we summarize them in Tab. 6. We mainly refer to the official code of GReg-1 (Wang et al., 2021b) when setting up these hyper-parameters, since we tap into a similar growing regularization scheme as GReg-1 does.

For small datasets (CIFAR and MNIST), each reported result is *averaged by at least 3 random runs*, mean and std reported. For ImageNet-1K, we cannot run multiple times due to our limited resource budget. This said, in general the results on ImageNet have been shown pretty stable.

---

[1]https://github.com/pytorch/examples/tree/master/imagenet

[2]https://github.com/MingSun-Tse/Regularization-Pruning

[3]https://github.com/samaonline/Orthogonal-Convolutional-Neural-Networks

[4]https://github.com/Eric-mingjie/rethinking-network-pruning/tree/master/imagenet/l1-norm-pruning

Table 5: Summary of training setups. In the parentheses of SGD are the momentum and weight decay. For LR schedule, the first number is initial LR; the second (in brackets) is the epochs when LR is decayed by factor 1/10; and #epochs stands for the total number of epochs.

| Dataset | MNIST | CIFAR10/100 | ImageNet |
|---|---|---|---|
| Solver | SGD (0.9, 1e-4) | SGD (0.9, 5e-4) | SGD (0.9, 1e-4) |
| Batch size | 100 | CIFAR10: 128, others: 256 | |
| LR schedule (scratch) | 1e-2, [30,60], #epochs:90 | 1e-1, [100,150], #epochs:200 | 1e-1, [30,60], #epochs:90 |
| LR schedule (prune) | Fixed (1e-3) | | |
| LR schedule (retrain) | 1e-2,[30,60], #epochs:90 | 1e-2, [60,90], #epochs:120 | 1e-2, [30,60,75], #epochs:90 |

Table 6: Hyper-parameters of our methods.

| Dataset | MNIST | CIFAR10/100 | ImageNet |
|---|---|---|---|
| Regularization granularity $\Delta$ | 1e-4 | | |
| Regularization ceiling $\tau$ | 1 | | |
| Regularization update interval $K_u$ | 10 iterations | | 5 iterations |

**Hardware and running time**. We conduct all our experiments using 4 NVIDIA V100 GPUs (16GB memory per GPU). It takes roughly 41 hrs to prune ResNet50 on ImageNet using our TPP method (pruning and 90-epoch retraining both included). Among them, 12 hrs (namely, close to 30%) are spent on pruning and 29 hrs are spent on retraining (about 20 mins per epoch).

**Layerwise pruning ratios**. The layerwise pruning ratios are *pre-specified* in this paper. For the ImageNet benchmark, we *exactly* follow GReg (Wang et al., 2021b) for the layerwise pruning ratios to keep fair comparison to it. The specific numbers are summarized in Tab. 7. Each number is the pruning ratio shared by all the layers of *the same stage* in ResNet34/50. On top of these ratios, some layers are skipped, such as the last CONV layer in a residual block. The best way to examine the detailed layerwise pruning ratio is to check the code at: https://github.com/MingSun-Tse/TPP.

Table 7: A brief summary of the layerwise pruning ratios (PRs) of ImageNet experiments.

| Model | Speedup | Pruned top-1 acc. (%) | PR |
|---|---|---|---|
| ResNet34 | 1.32× | 73.77 | [0, 0.50, 0.60, 0.40, 0, 0] |
| ResNet50 | 1.49× | 76.44 | [0, 0.30, 0.30, 0.30, 0.14, 0] |
| ResNet50 | 2.31× | 75.60 | [0, 0.60, 0.60, 0.60, 0.21, 0] |
| ResNet50 | 2.56× | 75.12 | [0, 0.74, 0.74, 0.60, 0.21, 0] |
| ResNet50 | 3.06× | 74.51 | [0, 0.68, 0.68, 0.68, 0.50, 0] |

## B  SENSITIVITY ANALYSIS OF HYPER-PARAMETERS

Among the three hyper-parameters in Tab. 6, regularization ceiling $\tau$ works as a termination condition. We only requires it to be large enough to ensure the weights are compressed to a very small amount. It does not have to be 1. The final performance is also less sensitive to it. The pruned performance seems to be more sensitive to the other two hyper-parameters, so here we conduct hyper-parameter sensitivity analysis to check their robustness.

Results are presented in Tab. 8 and Tab. 9. Pruning ratio 0.7 (for ResNet56) and 0.5 (for VGG19) are chosen here because the resulted sparsity is the most representative (*i.e.*, not too large or small). (1) For $K_u$, in general, a larger $K_u$ tends to deliver a better result. This is no surprise since a larger $K_u$ allows more iterations for the network to adapt and recover when undergoing the penalty. (2) For $\Delta$, we do not see a clear pattern here: either a small or large $\Delta$ can achieve the best result (for different networks). On the whole, when varying the hyper-parameters within a reasonable range, the performance is pretty robust, no catastrophic failures. Moreover, note that the default setting is actually not the best for both $K_u$ and $\Delta$. This is because we did *not* heavily search the best hyper-parameters; however, they still achieve encouraging performance compared to the counterpart methods, as we have shown in the main text.

Table 8: Robustness analysis of $K_u$ on the CIFAR10 and 100 datasets with our TPP algorithm. In default, $K_u = 10$. Layerwise PR = 0.7 for ResNet56 and 0.5 for VGG19. The best is highlighted in red and the worst in blue.

| $K_u$ | 1 | 5 | 10 | 15 | 20 |
|---|---|---|---|---|---|
| Acc. (%, ResNet56) | $92.24_{\pm0.03}$ | $92.42_{\pm0.14}$ | $92.35_{\pm0.12}$ | $92.50_{\pm0.10}$ | $92.31_{\pm0.16}$ |
| Acc. (%, VGG19) | $71.33_{\pm0.06}$ | $71.45_{\pm0.21}$ | $71.61_{\pm0.08}$ | $71.43_{\pm0.21}$ | $71.68_{\pm0.18}$ |

Table 9: Robustness analysis of $\Delta$ on the CIFAR10 and 100 datasets with our TPP algorithm. In default, $\Delta = 1e-4$. Layerwise PR = 0.7 for ResNet56 and 0.5 for VGG19. The best is highlighted in red and the worst in blue.

| $\Delta$ | 1e-5 | 5e-5 | 1e-4 | 5e-4 | 1e-3 |
|---|---|---|---|---|---|
| Acc. (%, ResNet56) | $92.37_{\pm0.12}$ | $92.29_{\pm0.10}$ | $92.35_{\pm0.12}$ | $92.40_{\pm0.15}$ | $92.44_{\pm0.13}$ |
| Acc. (%, VGG19) | $71.39_{\pm0.19}$ | $71.37_{\pm0.14}$ | $71.61_{\pm0.08}$ | $71.58_{\pm0.31}$ | $71.25_{\pm0.31}$ |

# C  ALGORITHM DETAILS

The details of our TPP method is summarized in Algorithm 1.

---
**Algorithm 1** Trainability Preserving Pruning (TPP)

---
1: **Input**: Pretrained model $\Theta$, layerwise pruning ratio $r_l$ of $l$-th layer, for $l \in \{1, 2, \cdots, L\}$.
2: **Input**: Regularization ceiling $\tau$, penalty coefficient update interval $K_u$, penalty granularity $\Delta$.
3: **Init**: Iteration $i = 0$. $\lambda_j = 0$ for all filter $j$. Set pruned filter indices $S_l$ by $L_1$-norm sorting.
4: **while** $\lambda_j \leq \tau$, for $j \in S_l$ **do**
5:     **if** $i \% K_u = 0$ **then**
6:         $\lambda_j = \lambda_j + \Delta$ for $j \in S_l$.    ▷ `Update regularization co-efficient in` `Eq. (`6`)`
7:     **end if**
8:     Network forward, loss (Eq. (6)) backward, parameter update by stochastic gradient descent.
9:     Update iteration: $i = i + 1$.
10: **end while**
11: Remove filters in $S_l$ to obtain a smaller model $\Theta'$.
12: Retrain $\Theta'$ to regain accuracy.
13: **Output**: Retrained model $\Theta'$.

---

# D  RESULTS OMITTED FROM THE MAIN TEXT

**Loss surface visualization**. The loss surface visualization of ResNet56 on CIFAR10 is presented in Fig. 4.

**VGG19 on CIFAR100**. The results of VGG19 on CIFAR100 is shown in Tab. 10.

**Examination of the early retraining phase**. To further understand how pruning hurts trainability and how our TPP method maintains it, in Tab. 11, we list the mean JSVs of the first 10-epoch retraining (at pruning ratio 0.9). Note the obvious mean JSV gap between LR 0.01 and LR 0.001 *without OrthP*: LR 0.01 can reach mean JSV 0.65 just after 1 epoch of retraining while LR 0.001 takes over 8 epochs. When OrthP is used, this gap greatly shrinks. We also list the test accuracy of the first 10-epoch retraining in Tab. 12. Particularly note that the *test accuracy correlates well with the mean JSV trend* under each setting, implying that the damaged trainability primarily answers for the under-performance of LR 0.001 after pruning.

Then, when TPP is used in place of OrthP, we can see after 1-epoch retraining, the model can achieve mean JSV above 1 and test accuracy over 90%, which are particularly similar to the effect of using OrthP. These re-iterate that the proposed TPP method can work just as effectively as the ground-truth trainability recovery method OrthP on this toy setup.

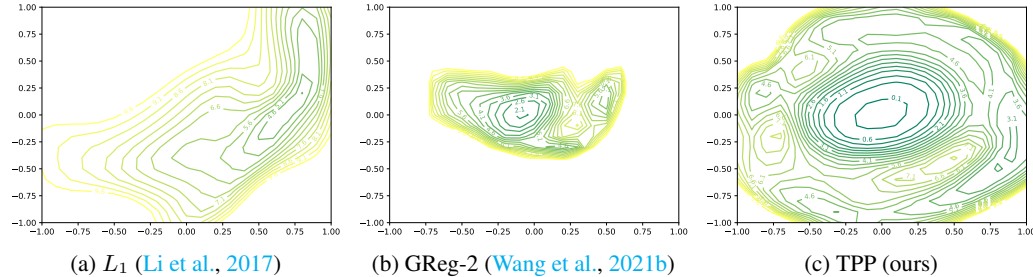

(a) $L_1$ (Li et al., 2017)      (b) GReg-2 (Wang et al., 2021b)      (c) TPP (ours)

Figure 4: Loss surface visualization of pruned models by different methods (w/o retraining). ResNet56 on CIFAR10. Pruning ratio: 0.9 (*zoom in to examine the details*).

Table 10: Test accuracy (%) comparison among different dynamical isometry maintenance or recovery methods on VGG19 on CIFAR100. *Scratch* stands for training from scratch. *KernOrth* means Kernel Orthogonalization (Xie et al., 2017); *OrthConv* means Convolutional Orthogonalization (Wang et al., 2020). Two retraining LR schedules are evaluated here: initial LR `1e-2` *vs.* `1e-3`. *Acc. diff.* refers to the accuracy gap of LR `1e-3` against LR `1e-2`.

| **VGG19 on CIFAR100**: Unpruned acc. 74.02%, Params: 20.08M, FLOPs: 0.80G | | | | | |
|---|---|---|---|---|---|
| Layerwise PR | 0.1 | 0.3 | 0.5 | 0.7 | 0.9 |
| Sparsity/Speedup | 19.24%/1.23× | 51.01%/1.97× | 74.87%/3.60× | 90.98%/8.84× | 98.96%/44.22× |
| | Initial | retraining | LR 1e-2 | | |
| Scratch | 72.84 (0.25) | 71.88 (0.14) | 70.79 (0.08) | 66.51 (0.11) | 54.37 (0.40) |
| $L_1$ (Li et al., 2017) | 74.01 (0.18) | 73.01 (0.22) | 71.49 (0.14) | 66.05 (0.04) | 51.36 (0.11) |
| $L_1$ + OrthP (Wang et al., 2021a) | 74.00 (0.04) | 72.30 (0.49) | 68.09 (0.24) | 62.22 (0.15) | 48.07 (0.31) |
| $L_1$ + KernOrth (Xie et al., 2017) | 73.72 (0.26) | 72.53 (0.09) | 71.23 (0.10) | 65.90 (0.14) | 50.75 (0.30) |
| $L_1$ + OrthConv (Wang et al., 2020) | 73.18 (0.10) | 72.25 (0.31) | 70.82 (0.11) | 64.51 (0.43) | 48.31 (0.18) |
| KernOrth (Xie et al., 2017) + $L_1$ | 73.73 (0.23) | 72.41 (0.12) | 70.31 (0.12) | 64.10 (0.19) | 50.72 (0.87) |
| OrthConv (Wang et al., 2020) + $L_1$ | 73.55 (0.18) | 72.67 (0.09) | 71.24 (0.23) | 65.66 (0.10) | 50.53 (0.46) |
| **TPP** (ours) | **74.02** (0.24) | **73.19** (0.07) | **71.61** (0.08) | **67.78** (0.31) | **57.70** (0.37) |
| | Initial | retraining | LR 1e-3 | | |
| $L_1$ (Li et al., 2017) | 73.67 (0.05) | 72.04 (0.12) | 70.21 (0.02) | 64.72 (0.17) | 48.43 (0.44) |
| **TPP** (ours) | **73.83** (0.02) | **72.29** (0.07) | **71.16** (0.12) | **67.47** (0.17) | **56.73** (0.34) |
| Acc. diff. ($L_1$) | -0.34 | -0.97 | -1.28 | -1.33 | -2.93 |
| Acc. diff. (TPP) | -0.19 | -0.90 | -0.45 | -0.31 | -0.97 |

# E   MORE ANALYTICAL RESULTS

In this section, we add more analytical results to help readers better understand how TPP works.

## E.1   COMPARISON UNDER SIMILAR TOTAL EPOCHS

In Tab. 1, TPP takes a few epochs for regularized training before the sparsifying action, while $L_1$ pruning is one-shot, taking no epochs. Namely, the total training cost of TPP is larger than those one-shot methods. It is of interest how the comparison will change if these one-shot methods are given more epochs for training.

First, note that the results of $L_1$+KernOrth, $L_1$+OrthConv, KernOrth+$L_1$, and OrthConv+$L_1$ in Tab. 1 also involve training (the KernOrth/OrthConv is essentially a regularized training), which takes 50k iterations. We make following changes:

- Our TPP takes 100k iterations based on the default hyper-parameter setup, so to make a fair comparison, we decrease the regularization update interval $K_u$ from 10 to 5, making the regularization of TPP also take 50k iterations.

- Meanwhile, we add 128 retraining epochs (50k / 391 iters per epoch ≈ 128 epochs) to the $L_1$ and $L_1$+OrthP methods (when their retraining epochs are increased, the LR decay epochs are proportionally scaled); plus the original 120 epochs, the total epochs are 248 now.

Table 11: Mean JSV of the first 10 epochs under different retraining settings. Epoch 0 refers to the model just pruned, before any retraining. Pruning ratio is 0.9. Note, with OrthP, the mean JSV is 1 because OrthP can achieve exact isometry

| Epoch | 0 | 1 | 2 | 3 | 4 | 5 | 6 | 7 | 8 | 9 | 10 |
|---|---|---|---|---|---|---|---|---|---|---|---|
| LR=$10^{-2}$, w/o OrthP | 0.00 | 0.65 | 0.89 | 1.01 | 0.97 | 1.10 | 1.14 | 1.27 | 1.33 | 1.29 | 1.42 |
| LR=$10^{-3}$, w/o OrthP | 0.00 | 0.00 | 0.00 | 0.00 | 0.10 | 0.25 | 0.35 | 0.42 | 0.51 | 0.74 | 0.86 |
| LR=$10^{-2}$, w/ OrthP | 1.00 | 1.23 | 1.42 | 1.43 | 1.40 | 1.47 | 1.51 | 1.56 | 1.60 | 1.65 | 1.68 |
| LR=$10^{-3}$, w/ OrthP | 1.00 | 1.50 | 1.64 | 1.73 | 1.84 | 1.87 | 1.93 | 1.96 | 1.99 | 2.00 | 2.04 |
| LR=$10^{-2}$, w/ TPP (ours) | 2.98 | 2.15 | 2.01 | 1.67 | 1.97 | 2.10 | 1.97 | 2.08 | 2.05 | 2.06 | 2.06 |
| LR=$10^{-3}$, w/ TPP (ours) | 2.98 | 2.96 | 2.95 | 2.99 | 2.99 | 3.01 | 3.21 | 3.19 | 3.05 | 3.04 | 3.02 |

Table 12: Test accuracy (%) of the first 10 epochs *corresponding to Tab. 11* under different retraining settings. Epoch 0 refers to the model just pruned, before any retraining. Pruning ratio is 0.9.

| Epoch | 0 | 1 | 2 | 3 | 4 | 5 | 6 | 7 | 8 | 9 | 10 |
|---|---|---|---|---|---|---|---|---|---|---|---|
| LR=$10^{-2}$, w/o OrthP | 9.74 | 64.34 | 80.01 | 80.23 | 81.77 | 85.80 | 85.82 | 86.21 | 86.35 | 86.60 | 86.15 |
| LR=$10^{-3}$, w/o OrthP | 9.74 | 9.74 | 9.74 | 11.89 | 21.34 | 27.75 | 32.96 | 35.38 | 49.66 | 64.89 | 68.97 |
| LR=$10^{-2}$, w/ OrthP | 9.74 | 91.05 | 91.39 | 91.33 | 91.37 | 91.74 | 91.69 | 90.74 | 91.39 | 91.58 | 91.44 |
| LR=$10^{-3}$, w/ OrthP | 9.74 | 90.81 | 91.59 | 91.77 | 91.85 | 92.04 | 92.12 | 92.22 | 92.12 | 92.33 | 92.25 |
| LR=$10^{-2}$, w/ TPP (ours) | 89.21 | 91.54 | 91.01 | 91.45 | 91.83 | 91.56 | 90.89 | 91.33 | 90.68 | 91.54 | 91.21 |
| LR=$10^{-3}$, w/ TPP (ours) | 89.21 | 92.12 | 91.82 | 92.09 | 92.15 | 91.95 | 92.00 | 92.02 | 92.09 | 92.08 | 92.08 |

Now all the comparison methods in Tab. 1 have the same training cost (*i.e.*, the same 248 total epochs). The new results of $L_1$, $L_1$+OrthP, and TPP under this strict comparison setup are presented below:

Table 13: Test accuracy comparison under the *same total epochs* (ResNet56 on CIFAR10).

| Layerwise PR | 0.3 | 0.5 | 0.7 | 0.9 | 0.95 |
|---|---|---|---|---|---|
| $L_1$ (Li et al., 2017) | 93.65 (0.14) | 93.38 (0.16) | 92.11 (0.16) | 87.17 (0.26) | 83.94 (0.45) |
| $L_1$ (Li et al., 2017)+OrthP | 93.58 (0.03) | 93.30 (0.10) | 91.69 (0.13) | 85.75 (0.26) | 82.30 (0.20) |
| TPP (ours) | 93.76 (0.10) | 93.45 (0.05) | 92.42 (0.14) | 89.54 (0.08) | 85.98 (0.29) |

We make the following observations:

- For $L_1$ pruning, more retraining epochs do not always help. Comparing these results to Tab. 1, we may notice at small PR (0.3, 0.5), the accuracy drops a little (this probably is due to overfitting – when the PR is small, the pruned model does not need so many epochs to recover; while too long training triggers overfitting). For larger PR (like 0.95), more epochs help quite significantly (improving the accuracy by 0.91%).

- $L_1$+OrthP still underperforms $L_1$, same as in Tab. 1.

- Despite using fewer epochs, TPP is still pretty robust – Compared to Tab. 1, the performance varies by a very marginal gap ( 0.1%, within the std range, so not a statistically significant gap). In general, TPP is still the best among all the compared methods, and, the advantage is more obvious at larger PRs, implying TPP is more valuable in more aggressive pruning cases.

## E.2 TPP + *Random* BASE MODELS

Pruning is typically conducted on a *pretrained* model. TPP is also brought up in this context. This said, the fundamental idea of TPP, *i.e.*, the proposed regularized training which can preserve trainability from the sparsifying action, is actually independent of the base model. Therefore, we may expect TPP can also surpass the $L_1$ pruning method (Li et al., 2017) in the case of pruning a *random* model – this is confirmed in Tab. 14. As expected, the damaged trainability problem does not

only exit for pruning pretrained models, but also exits for pruning random models; so TPP performs better than $L_1$ pruning, especially in the large sparsity regime.

Table 14: Test accuracy of applying TPP *vs.* $L_1$ pruning (Li et al., 2017) to *random base model*.

| **ResNet56 on CIFAR10**: Unpruned acc. 93.78%, Params: 0.85M, FLOPs: 0.25G | | | | | |
|---|---|---|---|---|---|
| Layerwise PR | 0.3 | 0.5 | 0.7 | 0.9 | 0.95 |
| $L_1$ pruning (Li et al., 2017) | 88.98 (0.13) | 88.79 (0.18) | 87.60 (0.21) | 85.09 (0.09) | 82.68 (0.32) |
| TPP (ours) | 91.93 (0.13) | 91.27 (0.16) | 90.36 (0.16) | 87.60 (0.09) | 85.36 (0.18) |

Moreover, in Fig. 5, we show the mean JSV and test accuracy when pruning a *random* model with different schemes ($L_1$, $L_1$+OrthP, and our TPP). We observe that, in general, the JSV and test accuracy of pruning a random model pose *similar* patterns to pruning a pretrained model (Fig. 3):

- Using $L_1$, the LR=0.01 achieves "better" (not really better, but due to damaged trainability, the performance of LR=0.001 has been underestimated, see Wang et al. (2023) for more detailed discussions) JSV and test accuracy; note the test accuracy solid line is above the dashed line by an obvious margin.

- While using $L_1$+OrthP or our TPP, the LR=0.001 can actually match LR=0.01. Same as the case of pruning a pretrained model, here, TPP behaves similarly to the oracle trainability recovery method OrthP.

- To summarize, the trainability-preserving effect of TPP also generalizes to the case of pruning random networks.

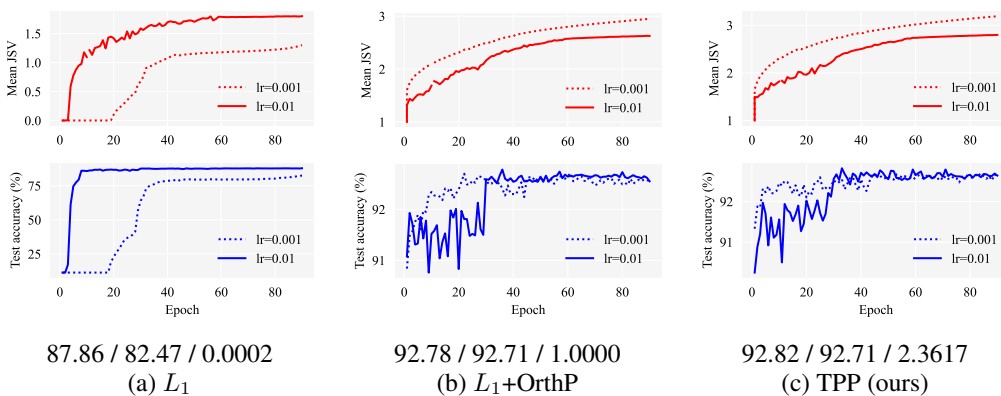

87.86 / 82.47 / 0.0002          92.78 / 92.71 / 1.0000          92.82 / 92.71 / 2.3617
     (a) $L_1$             (b) $L_1$+OrthP          (c) TPP (ours)

Figure 5: Mean JSV and test accuracy during *retraining* with different methods (network: MLP-7-Linear, dataset: MNIST) **when pruning a random model**. Below each plot are, in order, the best accuracy of LR `0.01`, the best accuracy of LR `0.001`, and the mean JSV right after pruning (*i.e.*, without retraining).

In Fig. 6, we show the mean JSV and test accuracy over the *overall process* (including the regularized training and retraining) of TPP applied to a random model.

### E.3   TPP + MORE ADVANCED PRUNING CRITERIA

In the main text, we demonstrate the effectiveness of TPP, which employs the *magnitude ($L_1$-norm)* of filters to decide masks. It is of interest whether the effectiveness of TPP can carry over to other more advanced pruning criteria. Here we combine TPP with other 3 more advanced criteria: SSL (Wen et al., 2016), DeepHoyer (Yang et al., 2020), and Taylor-FO (Molchanov et al., 2019). SSL and DeepHoyer are regularization-based pruning methods like ours; differently, their layerwise pruned indices (as well as the pruning ratio) is not pre-specified, but "learned" by the regularized training. As such, the layerwise pruning ratios are not uniform (see Fig. 7 for an example). Taylor-FO is a more complex pruning criterion than magnitude by taking into account the first-order gradient information.

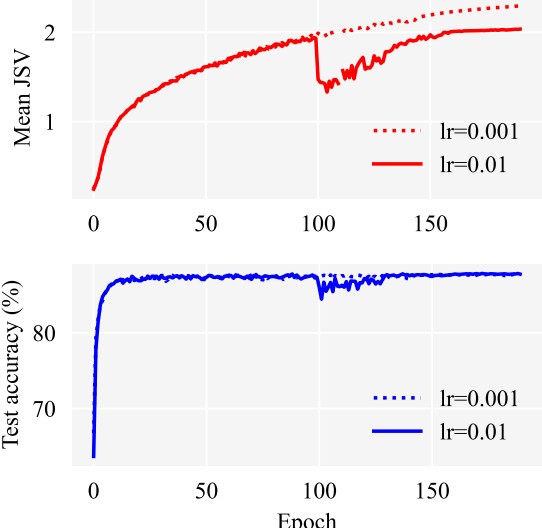

Figure 6: Mean JSV and test accuracy during *regularized training and retraining* with TPP (network: MLP-7-Linear, dataset: MNIST) **when pruning a random model**. The model spends 100 epochs on regularized training (LR schedule at this stage is the same, 0.001, fixed); then pruned by $L_1$ pruning (Li et al., 2017); then retrained with different LR schedules (0.01 *vs.* 0.001). As seen, pruning the *TPP regularized* model does *not* make the mean JSV drop significantly at the pruning point (epoch 100), namely, *trainability preserved*. In stark contrast, when applying $L_1$ pruning to a *normally trained* (*i.e.*, not TPP regularized) model, mean JSV drops from 2.4987 to 0.0040 (see Fig. 3), namely, trainability *not* preserved. Note, when trainability is preserved, retraining LR 0.01 and 0.001 do not pose obvious test accuracy gap; while trainability is not preserved, the gap would be obvious – see Fig. 5(a).

Specifically, we first use these methods to decide the layerwise pruned indices, given a total pruning ratio. Then, we inherit these layerwise pruned indices when using our TPP method.

Results are shown in Tab. 15. We observe, at small PRs (0.3-0.7), TPP performs similarly to $L_1$. This agrees with Tab. 1, where TPP is comparable to $L_1$. At large PRs (0.9, 0.95), the advantage of TPP starts to expose more – at PR 0.9/0.95, TPP beats $L_1$ by 0.9/1.44 with SSL learned pruned indices, which is a *statistically significant* advantage as indicated by the std (and again, when PR is larger, the advantage of TPP is generally more pronounced). This table shows the advantage of TPP indeed can carry over to other layerwise pruning ratios derived from more advanced pruning criteria.

### E.4    TRAINING CURVE PLOTS: TPP *vs.* $L_1$ PRUNING

In Fig. 8, we show the training curves of our TPP compared to $L_1$ pruning with ResNet56 on CIFAR10.

### E.5    PRUNING RESNET50 ON IMAGENET WITH LARGER SPARSITY RATIOS

It is noted that our method only beats some of the compared methods *marginally* ($<0.5\%$ top-1 accuracy) in low sparsity regime (around $2\times\sim3\times$ speedup, see Tab. 2). This is mainly because when the sparsity is low, network trainability is not seriously damaged, thus our *trainability*-preserving method cannot fully expose its advantage. Here we showcase a scenario that trainability is intentionally damaged more dramatically.

In Tab. 2, when pruning ResNet50, researchers typically do not prune all the layers – the last CONV layer in a residual block is usually spared (Li et al., 2017; Wang et al., 2021b), for the sake of performance. Here we intentionally prune *all* the layers (only excluding the first CONV and last

Table 15: Comparison between $L_1$ pruning (Li et al., 2017) and our TPP with pruned indices derived from more advanced pruning criterion (Taylor-FO (Molchanov et al., 2019)) or regularization schemes (SSL (Wen et al., 2016), DeepHoyer (Yang et al., 2020)). Network/Dataset: ResNet56/CIFAR10: Unpruned acc. 93.78%, Params: 0.85M, FLOPs: 0.25G. *Total PR* represents the pruning ratio (PR) of the whole network. Note, due to the non-uniform layerwise PRs, the speedup below, which depends on the feature map spatial size, can be quite different from each other, even under the same total PR.

| Total PR | 0.3 | 0.5 | 0.7 | 0.9 | 0.95 |
|---|---|---|---|---|---|
| Sparsity/Speedup | 22.86%/1.66× | 47.09%/2.25× | 72.28%/3.32× | 92.49%/7.14× | 95.87%/9.77× |
| $L_1$ (w/ SSL pruned indices) | $93.87_{\pm 0.02}$ | $93.47_{\pm 0.04}$ | $92.76_{\pm 0.15}$ | $89.00_{\pm 0.13}$ | $84.75_{\pm 0.21}$ |
| TPP (w/ SSL layerwise pruned indices) | $93.86_{\pm 0.09}$ | $93.50_{\pm 0.15}$ | $92.81_{\pm 0.05}$ | $89.90_{\pm 0.07}$ | $86.19_{\pm 0.22}$ |
| Sparsity/Speedup | 26.87%/1.54× | 52.11%/1.95× | 76.50%/2.74× | 93.21%/6.15× | 96.02%/9.60× |
| $L_1$ (w/ DeepHoyer pruned indices) | $93.81_{\pm 0.09}$ | $93.81_{\pm 0.10}$ | $92.33_{\pm 0.06}$ | $87.61_{\pm 0.15}$ | $84.27_{\pm 0.10}$ |
| TPP (w/ DeepHoyer pruned indices) | $93.88_{\pm 0.13}$ | $93.59_{\pm 0.04}$ | $92.58_{\pm 0.20}$ | $88.76_{\pm 0.17}$ | $85.64_{\pm 0.18}$ |
| Sparsity/Speedup | 31.21%/1.45× | 49.94%/1.99× | 70.75%/3.59× | 90.66%/11.58× | 95.41%/19.85× |
| $L_1$ (w/ Taylor-FO pruned indices) | $93.91_{\pm 0.11}$ | $93.50_{\pm 0.05}$ | $92.24_{\pm 0.12}$ | $87.28_{\pm 0.32}$ | $83.31_{\pm 0.43}$ |
| TPP (w/ Taylor-FO pruned indices) | $93.91_{\pm 0.09}$ | $93.64_{\pm 0.14}$ | $92.32_{\pm 0.13}$ | $88.06_{\pm 0.10}$ | $85.34_{\pm 0.32}$ |

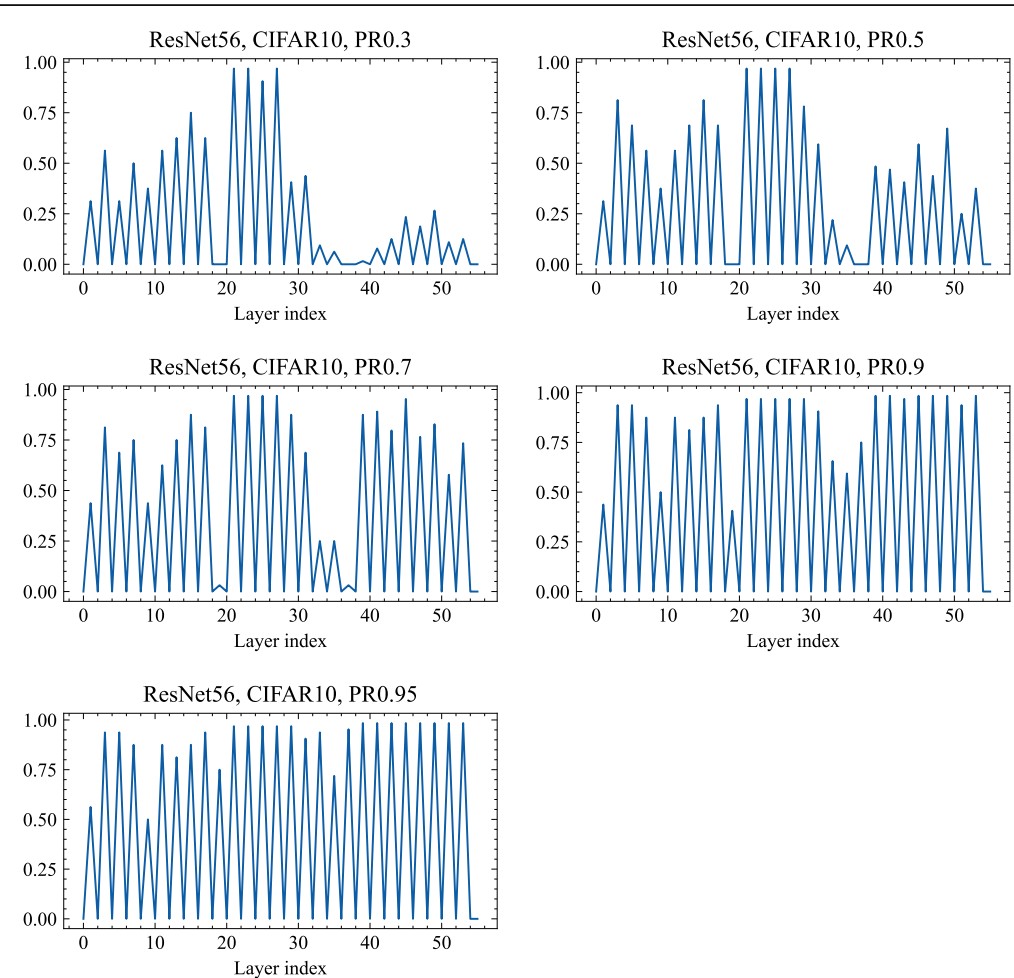

Figure 7: Layerwise pruning ratios learned by SSL (Wen et al., 2016) with ResNet56 on CIFAR10, given different *total pruning ratios* (indicated in the title of each sub-figure).

classifier FC) in ResNet50. For the $L_1$ pruning method (Li et al., 2017) (we report a stronger version re-implemented by Wang et al. (2023)), different layers are pruned independently since the layerwise pruning ratio has been specified. All the hyper-parameters of the retraining process are maintained *the same* for fair comparison, per the spirit brought up in Wang et al. (2023).

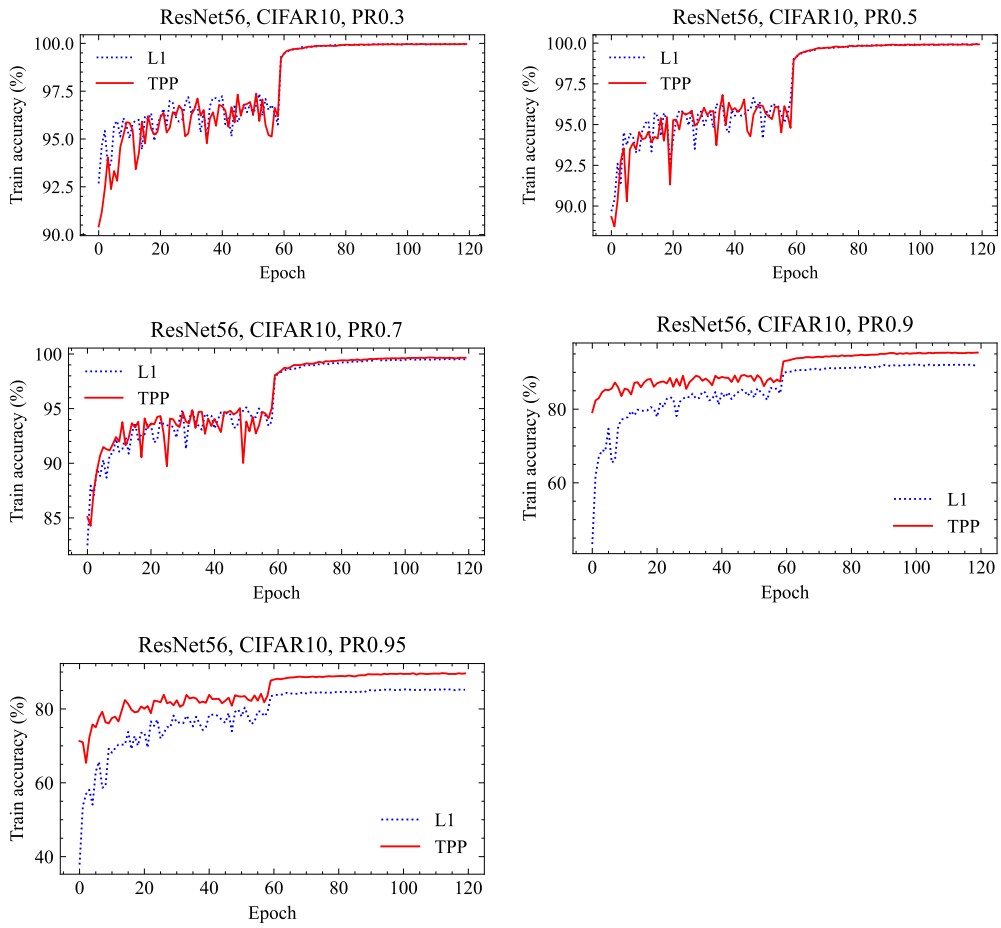

Figure 8: Training curves during retraining with ResNet56 on CIFAR10 at different pruning ratios (PRs). We can observe that at large PRs (0.9, 0.95), TPP significantly accelerates the optimization in the comparison to $L_1$ (Li et al., 2017), because of better trainability preserved before retraining.

The results in Tab. 16 show that, when the trainability is impaired more, our TPP beats $L_1$ by 0.77 to 2.17 top-1 accuracy on ImageNet, much more significant than Tab. 2.

Table 16: Top-1 accuracy comparison between TPP and $L_1$ pruning with larger pruning ratios (PRs). All layers but the 1st CONV and last FC layer (including the downsample layers and the 3rd CONV in a residual block) are pruned. Uniform layerwise pruning ratio is used.

| Layerwise PR
Sparsity/Speedup | 0.5
72.94%/3.63× | 0.7
89.34%/8.45× | 0.9
98.25%/25.34× | 0.95
99.34%/31.45× |
|---|---|---|---|---|
| $L_1$ (Li et al., 2017)
TPP (ours) | 71.25
73.42 (+2.17) | 66.02
68.16 (+2.14) | 47.96
49.19 (+1.23) | 33.21
33.98 (+0.77) |

## F    MORE DISCUSSIONS

*Can TPP be useful for finding lottery ticket subnetwork in a filter level?*

To our best knowledge, filter-level winning tickets (WT) are still hard to find even using the original LTH pipeline. Few attempts in this direction have succeeded – *E.g.*, Chen et al. (2022) tried, but they can only achieve a bit marginal sparsity ( 30%) with filter-level WT (see their Fig. 3, ResNet50 on ImageNet) while weight-level WT typically can be found at over 90% sparsity. This said, we do think this paper can contribute in that direction since preserving trainability is also a central issue in LTH, too.

