# OpenReview forum: "Trainability Preserving Neural Pruning"
_ICLR.cc/2023/Conference — ICLR 2023 poster_

### Official Review · Reviewer_LVfu · 2022-10-23

**Confidence:** 5
**Correctness:** 3
**Technical Novelty And Significance:** 3
**Empirical Novelty And Significance:** 3
**Recommendation:** 6

**Clarity, Quality, Novelty And Reproducibility:**

The paper makes novel and significant contribution by introducing the concept of trainability preservation into the neural network pruning process. The paper is overall clearity written and easy to follow. The quality of the paper is generally good, but can be improved by further showing the model behavior during the regularized training phase with the proposed method, and consider the additional training steps needed by the proposed method in discussing the experiment results.

**Strength And Weaknesses:**

## Strength
1. The paper makes novel and significant contribution by introducing the concept of trainability preservation into the neural network pruning process.
2. The proposed method is well motivated and theortically sound
3. Thorough emperical evaluation is provided to prove the effectiveness of the proposed method

## Weakness
1. The paper only demonstrates the model performance during the finetuning process. As a regualrization paper, it would be interesting to show how the regularization perform during the regularized training process. Things like how does the regularization affect model accuracy before pruning, and how it changes the mean JSV. I would encourage the author to repeat Figure 3 with the regualrized training epochs before the pruning.
2. To my understanding the regularizer requires additional training epochs before pruning to take effect, however there's no discussion on how many epochs is needed, and how it's decided. This may also cause unfair comparison for the results in Table 1 and 2, as some baselines may be trained with less epochs in total (both before and after pruning), and may benefit from more training epochs
3. The paper limits the exploration on L1 megnitude-based pruning with fixed pruning precentage on each layer. However there are more advanced filter pruning techniques like Taylor importance-based global pruning[1], or dynamically determining the filters to be pruned during regularized training process[2,3]. It would be interesting to see if the proposed method works under these pruning schemes.

[1] https://arxiv.org/abs/1906.10771

[2] https://proceedings.neurips.cc/paper/2016/file/41bfd20a38bb1b0bec75acf0845530a7-Paper.pdf

[3] https://arxiv.org/pdf/1908.09979.pdf

**Summary Of The Paper:**

This paper propose a regularization method to maintain the trainability of the model during the filter pruning process. The regularization is applied on both the weight and the BN parameters. Experiment results on multiple dataset and model architectures show promising performance comparing to other SOTA pruning methods.

**Summary Of The Review:**

Generally the paper proposes an interesting and strong method for improving the finetuning trainability of the pruned model. The method is novel and solid. Meanwhile there are some concern on the behavior of the model during the regularized training phase and the fairness of the experiment. I would recommend a weak acceptance for now, expecting the author to clear my doubt in the rebuttal.

## Post rebuttal

I have read other reviews and all responses from the author. Most of my concerns are resolved. The paper provides a novel prespective for improving DNN pruning, where is shows small but consistent improvement in accuracy under practical sparsity, and larger accuracy improvement under high sparsity. I would keep my score as weak accept.

---

> ### Author Response · Authors · 2022-11-16
> **Thank you! Responses to Reviewer LVfu (Part2)**
>
> > `Q3`: The paper limits the exploration on L1 megnitude-based pruning with fixed pruning precentage on each layer. However there are more advanced filter pruning techniques like Taylor importance-based global pruning[1], or dynamically determining the filters to be pruned during regularized training process[2,3]. It would be interesting to see if the proposed method works under these pruning schemes.
>
> `A3`: Thank you for pointing out this issue! We fully understand your concern if our method can still work for more pruning criteria other than the L1-norms, especially those derived by regularization (like SSL [2], DeepHoyer [3]) or more advanced criteria (like Taylor-FO [1]). As suggested, we integrate our method with the three methods you suggested: SSL, DeepHoyer, and Taylor-FO. Specifically, we use these methods to *learn* the layerwise pruned filter indices (the layerwise pruning ratio is thus also learned instead of predefined) and then apply our TPP method with these pruned indices.
>
> The results of our TPP vs. "L1" based on these pruned indices are presented below (note, due to the non-uniform layerwise PRs, the speedup below, which depends on the feature map spatial size, can be quite different from those in Tab. 1 of our paper, even under the same total PR, e.g., total PR = 0.95):
> | Total PR | 0.3 | 0.5 | 0.7 | 0.9 | 0.95 |
> |---|---|---|---|---|---|
> | Sparisty/Speedup | 22.86%/1.66x | 47.09%/2.25x | 72.28%/3.32x | 92.49%/7.14x | 95.87%/9.77x |
> | L1 (w/ SSL layerwise pruned indices) | 93.87$_{\pm0.02}$ | 93.47$_{\pm0.04}$ | 92.76$_{\pm0.15}$ | 89.00$_{\pm0.13}$ | 84.75$_{\pm0.21}$ |
> | TPP (w/ SSL layerwise pruned indices) | 93.86$_{\pm0.09}$ | 93.50$_{\pm0.15}$ | 92.81$_{\pm0.05}$ | 89.90$_{\pm0.07}$ | 86.19$_{\pm0.22}$
>
> | Total PR | 0.3 | 0.5 | 0.7 | 0.9 | 0.95 |
> |---|---|---|---|---|---|
> | Sparisty/Speedup | 26.87%/1.54x | 52.11%/1.95x | 76.50%/2.74x | 93.21%/6.15x | 96.02%/9.60x |
> | L1 (w/ DeepHoyer layerwise pruned indices) | 93.81$_{\pm0.09}$ | 93.81$_{\pm0.10}$ | 92.33$_{\pm0.06}$ | 87.61$_{\pm0.15}$ | 84.27$_{\pm0.10}$ |
> | TPP (w/ DeepHoyer layerwise pruned indices) |93.88$_{\pm0.13}$ | 93.59$_{\pm0.04}$ | 92.58$_{\pm0.20}$ | 88.76$_{\pm0.17}$ | 85.64$_{\pm0.18}$
>
> | Total PR | 0.3 | 0.5 | 0.7 | 0.9 | 0.95 |
> |---|---|---|---|---|---|
> | Sparisty/Speedup | 31.21%/1.45x | 49.94%/1.99x | 70.75%/3.59x | 90.66%/11.58x | 95.41%/19.85x |
> | L1 (w/ Taylor-FO layerwise pruned indices) | 93.91$_{\pm0.11}$ | 93.50$_{\pm0.05}$ | 92.24$_{\pm0.12}$ | 87.28$_{\pm0.32}$ | 83.31$_{\pm0.43}$
> | TPP (w/ Taylor-FO layerwise pruned indices) | 93.91$_{\pm0.09}$ | 93.64$_{\pm0.14}$ | 92.32$_{\pm0.13}$ | 88.06$_{\pm0.10}$ | 85.34$_{\pm0.32}$ |
>
> **Observations**:
> - At small PRs (0.3-0.7), TPP performs similarly to L1. This agrees with Tab. 1, where TPP is comparable to L1. At large PRs (0.9, 0.95), the advantage of TPP starts to expose more -- at PR 0.9/0.95, TPP beats L1 by 0.9/1.44 with SSL learned pruned indices, which is a *statistically significant* advantage as indicated by the std (and again, when PR is larger, the advantage of TPP is generally more pronounced).
> - This table shows the advantage of TPP indeed can *carry over to other layerwise pruning ratios* derived from more advanced pruning criteria.
>
> We will cite these papers and include the results in our final version.
>
> ---
> **Lastly, *thank you so much* for helping us improve the paper! Please let us know if you have *any* further questions. We are *actively available* until the end of this rebuttal period. Looking forward to hearing back from you!**

---

> > ### Comment · Reviewer_LVfu · 2022-11-16
> > **Feedback on author response**
> >
> > I would like to thank the author for the responses. The response settles my second concern. Looking forward to additional results on the other points.

---

> > > ### Author Response · Authors · 2022-12-04
> > > **All suggested results by Reviewer LVfu have been updated, thank you!**
> > >
> > > Dear reviewer LVfu,
> > >
> > > Thank you for helping us improve the paper so far! We just finished *all* the experiments as you suggested. As seen, **the effectiveness of TPP indeed carries over to more advanced pruning criteria** (with non-uniform layerwise pruning ratios learned by regularization or decided by Hessian criteria),  especially at large sparsity regimes.
> > >
> > > We were wondering, given your concerns are resolved, would you generously consider raising the score even a little bit as a fair recognition of our rebuttal efforts so far? Thank you so much!
> > >
> > > Sincerely,
> > >
> > > Authors

---

> > > > ### Comment · Reviewer_LVfu · 2022-12-06
> > > > **Thanks for the response**
> > > >
> > > > I would like to thank the author for the updated experiments. The new reseults meet my expectation, and proves the effectiveness of the proposed method. However, the effectiveness of the proposed method mainly appears at high sparsity ratio, which may not be practically useful in real applications (similar point is also made by reviewer Qf3U).
> > > >
> > > > Nevertheless, I do think the paper makes novel contribution on introducing the concept of trainability preservation into the neural network pruning process, and providing an effective regularization to preserve trainablity. Though the performance improvement may be small under practical sparsity, the consistent improvement also indicates a positive contribution. Balancing the pros and cons, I would like to keep my score and advocate for acceptance.

---

> > > > > ### Author Response · Authors · 2022-12-06
> > > > > **Thank you!**
> > > > >
> > > > > Thank you reviewer LVfu! We really appreciate your feedback!

---

> ### Author Response · Authors · 2022-11-16
> **Thank you! Responses to Reviewer LVfu (Part 1)**
>
> Dear Reviewer LVfu,
>
> ***Thank you so much*** for your very valuable comments and the positive score! We address your concerns as follows.
>
> > `Q1`: The paper only demonstrates the model performance during the finetuning process. As a regualrization paper, it would be interesting to show how the regularization perform during the regularized training process. Things like how does the regularization affect model accuracy before pruning, and how it changes the mean JSV. I would encourage the author to repeat Figure 3 with the regualrized training epochs before the pruning.
>
> `A1`: Thanks for letting us know about this concern. Your point is correct -- TPP can be directly used in the regularized training process (vs. regularizing a pretrained model). Reviewer p5ry also has this concern. The results of employing TPP under this setup are presented in `Responses to Reviewer p5ry (Part 1), A2`, which you may refer to there.
>
> Meanwhile, per your suggestion, we report the results (accuracy, JSVs) before and after pruning when TPP is used for regularizing a *random* model, shown in [this link (after pruning)](https://imgur.com/a/lUxyA9X) (also included in the revised pdf, see Appendix Fig. 5 and 6).
>
> **Observations**: The JSV and test accuracy of pruning a random model pose *similar patterns* to pruning a pretrained model (Fig. 3):
> - Using "L1", the LR=0.01 achieves  "better" (not really better, but due to damaged trainability, the performance of LR=0.001 has been underestimated) JSV and test accuracy; note the test accuracy solid line is above the dashed line by an obvious margin.
> - While using "L1+OrthP" or our "TPP", the LR=0.001 can actually *match* LR=0.01. Same as the case of pruning a pretrained model, here, TPP behaves similarly to the oracle trainability recovery method OrthP.
> - To summarize, the trainability-preserving effect of TPP also generalizes to the sparse training case.
>
> ---
> > `Q2`:To my understanding the regularizer requires additional training epochs before pruning to take effect, however there's no discussion on how many epochs is needed, and how it's decided. This may also cause unfair comparison for the results in Table 1 and 2, as some baselines may be trained with less epochs in total (both before and after pruning), and may benefit from more training epochs
>
> `A2`: "*however there's no discussion on how many epochs is needed, and how it's decided.*" -- The epochs can be derived from the hyper-parameters we list in the Appendix (see Tab. 6). The regularization ceiling is $\tau$, update interval is $K_u$, granularity is $\Delta$, so the needed iterations for the TPP regularization is $\tau/\Delta * K_u$. $\Delta$ is decided by *referring to the weight decay* (since the TPP regularization is essentially a variant of L2 regularization like weight decay) on ImageNet (i.e., 0.0001) . $\tau = 1$ is empirically decided, which only needs to be a large penalty number and does not have to be 1. $K_u$ is also empirically decided. The sensitivity analysis of $K_u$ has been shown in Tab. 8 of the Appendix. *Please refer to Appendix A.2 for more details about how we decide these hyper-parameters*.
>
> "*This may also cause unfair comparison for the results in Table 1 and 2, as some baselines may be trained with less epochs in total (both before and after pruning), and may benefit from more training epochs*" --
> - About the comparison fairness in Tab. 1, Reviewer p5ry also has this concern. To resolve this concern, we augment the other methods ("L1" and "L1 + OrthP"; the other 4 methods do not need reruns because they already meet the epochs requirement) with more epochs (128 epochs); and meanwhile, we halve the training epochs of TPP. This way, *all the methods will spend ~248 epochs in total*, thus the comparison under this setup would be *strictly fair*. The new results under this setup are presented in the `Responses to Reviewer p5ry (Part 2), A3`. You may see them there.
> - For Tab. 2, for one thing, it is on ImageNet (which is quite costly for our computation resources); for another thing, it is pretty hard to reimplement other methods (many papers do not even release their code), so we could not have new results at a strictly fair setup as we do above on the CIFAR dataset. Yet note, based on our hyper-parameter settings (see Appendix Tab. 6), the TPP regularization process on ImageNet only takes $1/0.0001*5 / 5005 = 10$ epochs (5005 is the number of iterations per epoch on ImageNet with batch size 256), which is *not a large number of epochs* considering the finetuning typically spends *many more* (e.g., CCP-AC (Peng et al., 2019) spends 100 finetuning epochs; GReg (Wang et al., 2021b) spends 90; Factorized (Li et al., 2019) spends 90). Therefore, we think the results on ImageNet in Tab. 2 are still fair in general.
>
> Thank you for letting us know that we actually did not explain these clearly in the paper. We would add the new results and these discussions in the revised version to clarify this point.

---

### Official Review · Reviewer_Qf3U · 2022-10-24

**Confidence:** 4
**Correctness:** 3
**Technical Novelty And Significance:** 3
**Empirical Novelty And Significance:** 2
**Recommendation:** 6

**Clarity, Quality, Novelty And Reproducibility:**

* The paper is presented clearly, with sufficient related works and related discussions.
* Experiments come with sufficient details. Training parameters are provided in appendix. It should not be difficult to reproduce results in this work.
* This work lacks novelty in both theory and methodology. The practical impact may be very limited.

**Strength And Weaknesses:**

[Strength]

* The paper is overall well-written and easy to follow.
* The authors compared their method to multiple strong baseline methods, such as OrthConv and KernOrth.
* In the finetuning, the authors compared two different learning rates. All methods are trained under the same settings for fairness.
* Experiment results on ImageNet are reported, in addition to the CIFAR-10/100, including a dozen baseline methods.
* The accuracy improvement is significant on CIFAR-10/100

[Weakness]
* The novelty is incremental. Orthogonal regularization is not new in training deep network. Dating back to years ago, there were lots of discussions of weight regularization to make network more trainable, such as dynamic isometry constraint or nuclear norm regularizer. This work extends these ideas to network pruning, with some heuristic adaptations. For example, the correlation between selected important filters is not penalized in the proposed regularizer. BN norms of unimportant filters are considered. However, all these adaptations lack strong theoretical motivations. The authors did not provide any theoretical insight of why the proposed decorrelation plays a critical role in deep neural network pruning. From the numerical results, it seems that the improvement of the proposed method is also marginal on large-scale datasets.

* Marginal improvement on large-scale datasets such as ImageNet. The difference sof the proposed method v.s. previous SOTA methods are around or less than 0.5% in most cases. This is not significant, especailly when the training setting in this work is not SOTA. For example, using a better training recipe in "Resnet strikes back: An improved training procedure in timm“, it is possible to improve ResNet-50 from 76% to 80% on ImageNet-1k. The 0.5% improvement is hardly to say a significant number.

* No trained long enough. In SOTA training setting, ResNet-50 on ImageNet requires 360 epochs of training. Training even longer can still improve accuracy a bit. In this paper, most training experiments are early-stopped. So it is hard to tell the final accuracies when all models are trained by 360 epoches.

* Training curves not reported. As the key argument of this work is trainability, the convergence curve should be plotted. Again, please consider to train long enough such that all models achive their stable convergent points. Then please compare the convergence speed.


**Summary Of The Paper:**

The authors propose a neural network pruning method which perserves trainability after pruning. The authors argue that the proposed approach will make the network easier to train after pruning therefore lead to a better finetuned model. First, the proposed method use L1 norm to select important and unimportant filters. Then,  a trainability regularization term is induced during the finetuning step which enforces the important filters to be orthogonal to unimportant ones. In addition, parameters in BN layers are regularized by L2 norm for those unimportant filters. The authors validate the proposed method on CIFAR-10/100 and ImageNet-1k.

**Summary Of The Review:**

The majority concern is lacking novelty. Some improvments in the experiments would make this work more stronger and more convincing.

---

> ### Author Response · Authors · 2022-11-15
> **Thank you! Responses to Reviewer Qf3U (Part 2)**
>
> > `Q2`: Marginal improvement on large-scale datasets such as ImageNet. The difference sof the proposed method v.s. previous SOTA methods are around or less than 0.5% in most cases. This is not significant, especailly when the training setting in this work is not SOTA. For example, using a better training recipe in "Resnet strikes back: An improved training procedure in timm“, it is possible to improve ResNet-50 from 76% to 80% on ImageNet-1k. The 0.5% improvement is hardly to say a significant number.
>
> `A2`: "*The difference sof the proposed method v.s. previous SOTA methods are around or less than 0.5% in most cases.*" -- This is mainly because of two reasons -- (1) the speedup ratios in Tab. 2 are not very large (the greatest speedup is only ~3x; we choose this range mainly for maintaining comparison with prior works; it is the speedup range that most past works reported) -- it is well known that when the speedup is small, the difference between different methods is small too -- this can be seen from Tab. 1 where the advantage of TPP is more pronounced at pruning ratio 95% than pruning ratio 30%; (2) And note, what we compare to are the very recent SOTA methods. It is non-trivial to beat so many top-performing methods.
>
> " *especailly when the training setting in this work is not SOTA. For example, using a better training recipe in "Resnet strikes back: An improved training procedure in timm“, it is possible to improve ResNet-50 from 76% to 80% on ImageNet-1k. The 0.5% improvement is hardly to say a significant number."* " -- Regarding this concern,
> - first, we are aware of the TIMM training recipe, and actually, we *have presented* such results using the advanced TIMM training recipe, **see Tab. 2 (the last two rows)**.
> - second, since most comparison methods in Tab. 2 simply use the old training recipe (standard data augmentation: crop+flip, step decay LR with ~90 finetuning epochs), if we use TIMM (which integrates *a bag of tricks* of advanced data augmentation, cosine LR, long epochs like 300 epochs), yes, definitely, we will achieve high performance with more than 0.5% advantage (and we have achieved it -- see the last two rows of Tab. 2), but it would be *obviously unfair* if we compare these results to other pruning methods that only use the common training recipe. This is why we single out the results using TIMM in Tab. 2 in the last two rows.
>
> ---
> > `Q3`: No trained long enough. In SOTA training setting, ResNet-50 on ImageNet requires 360 epochs of training. Training even longer can still improve accuracy a bit. In this paper, most training experiments are early-stopped. So it is hard to tell the final accuracies when all models are trained by 360 epoches.
>
> `A3`: This concern has been addressed in `A2` -- (1) We have presented the results using TIMM which trains for 300 epochs (see the last two rows of Tab. 2). (2) More importantly, high performance is not the only thing we are after; we value more the *comparison fairness* -- since most existing pruning methods did *not* use such advanced training recipe, *to keep a fair comparison*, the major results in Tab. 2 are still using the common 90-epoch training recipe.
>
> ---
> > `Q4`: Training curves not reported. As the key argument of this work is trainability, the convergence curve should be plotted. Again, please consider to train long enough such that all models achive their stable convergent points. Then please compare the convergence speed.
>
> `A4`: Thank you for pointing this out! We agree this should be reported. We report the training curves with ResNet56 on CIFAR10 under various pruning ratios (PRs): see [this link](https://imgur.com/a/27HAb62) (also included in the revised pdf, see Appendix Fig. 8)
>
> **Observations**: As seen, at a small PR (0.3 - 0.7), the convergence speed and accuracy gap are only marginal, while at a large PR (0.9, 0.95), the accuracy gap is pretty obvious. This is in line with the performance gap shown in Tab. 3 -- the advantage of TPP is more pronounced at large PRs (as at large PRs, the network trainability is damaged more, where TPP can find more use in these cases).
>
> "*Again, please consider to train long enough such that all models achive their stable convergent points*" -- we have reported such results on ImageNet in Tab. 2 (see the last two rows).
>
> ---
> **Lastly, *thank you so much* for helping us improve the paper! Please let us know if you have *any* further questions. We are *actively available* until the end of this rebuttal period. Looking forward to hearing back from you!**

---

> > ### Comment · Reviewer_Qf3U · 2022-11-21
> > **Thanks for the authors' feedback**
> >
> > I still have concerns after reading authors' feedback.
> >
> > 1) The theoretical contribution is very limited and marginal, as argued in the initial review. Please note that I did not vote for reject simply because lack of theoretical contribution. If the paper shows strong empirical improvements, I would be very happy to accept it. Therefore, I separate the contributions of this work into two parts: theoretical part and empirical part.
> >
> > 2) From theoretical side, this work applied weight orthgonal regularizer to pruning, so a) it is an incremental idea; b) it did not show more insight on why the orthgonal regularizer, in theory, can lead to better pruning algorithm. The connection is not very clear.
> >
> > 3) From empirical side, the improvments of the proposed method on ImageNet-1k is not very significant. As the authors also agree, the model accuracy should be compared under fair training setting, otherwise it is meaningless, because different training settings can lead to dramatically different accuracies. In Table 2, it seems that only CHEX and the last two rows are trained using the same TIMM SOTA training settings.
> >
> > 4) Thanks for the confirmation of training settings used in the last two rows of Table 2. Yet, the improvements are still around 0.5%, hardly to say significant to me.
> >
> > 5) The convergence curve seems interesting and thanks for the update! On CIFAR-10, when the pruning ratio is 30%~70%, the naive L1-norm pruning shows the same curve as the proposed method. This does not support the key argument of this work. Also, what about the other more advanced pruning methods?
> >
> > 6) Again, in the convergence curve figure, when pruning ratio is 90%, the proposed method is indeed faster and better than the naive L1-norm pruning. However, at this sparsity level, the accuracy degrades a lot, meaning that the not only the redundant neurons but also informative neurons are pruned at this sparsity level. The practical value of this setting is in question (although I did not deny its value completely).  Also, what about the other more advanced pruning methods under this setting?

---

> > > ### Author Response · Authors · 2022-11-21
> > > **Thanks for your followup! (Part 2)**
> > >
> > > > `Q5 (2)`: "*Also, what about the other more advanced pruning methods?*"
> > >
> > > `A5 (2)`: Reviewer LVfu also has this concern: Can the findings generalize to other more advanced pruning methods or criteria? They named 3 methods for us to try (see `Responses to Reviewer LVfu (Part2), A3`), you may refer to them first for the accuracy results (again, you'll see at low sparsities, the performance gap is not obvious). Meanwhile, we'll add the training curves when these advanced methods are used. When we have more results, we will update here. Stay tuned. Thanks!
> > >
> > > ---
> > > > `Q6 (1)`: "*Again, in the convergence curve figure, when pruning ratio is 90%, the proposed method is indeed faster and better than the naive L1-norm pruning. However, at this sparsity level, the accuracy degrades a lot, meaning that the not only the redundant neurons but also informative neurons are pruned at this sparsity level. The practical value of this setting is in question (although I did not deny its value completely).*"
> > >
> > > `A6 (1)`: Thank you for letting us know about this and thank you for recognizing the value! We fully understand your concern. We present such cases (extreme pruning ratios like 90%/95%) simply to give a *comprehensive* view of the method's behavior (many pruning papers only show the performance at a few sparsities, not a full-spectrum of sparsity, which may miss some corner cases). These are more for analysis instead of for practical usage (in fact, we use *uniform* PRs for all our CIFAR/MNIST experiments, which are rarely used in practice if performance is our target, e.g., in Tab. 2).
> > >
> > > To see the *"practical value"* of a pruning method, people usually check out the ImageNet results. In this regard, see Tab. 2, where we have shown TPP performs favorably against many methods, including the very recent top-performing methods. In our updated results (see `Thanks for your followup! (Part 1), A3/4`), we show at greater speedups, our TPP indeed delivers **significant** performance improvement.
> > >
> > > Lastly, the value/performance of a pruning method is always a *tradeoff* -- better performance but more parameters/cost or degraded performance but fewer parameters/cost. When sparsity goes beyond some point (i.e., the redundancy point), we would always see the performance drop. So the point when evaluating a pruning method is not how the method performs *against its unpruned model*, but *against other pruning methods* at the same sparsity. In this sense, note our method is obviously better than L1, as acknowledged by Reviewer Qf3U.
> > >
> > > ---
> > > > `Q6 (2)`: "*Also, what about the other more advanced pruning methods under this setting?*"
> > >
> > > `A6 (2)`: See `Responses to Reviewer LVfu (Part2), A3`. We try 3 more advanced pruning methods suggested by Reviewer LVfu. The effectiveness of our method can carry over to those cases. We shall include these results in our final version.

---

> > > ### Author Response · Authors · 2022-11-21
> > > **Thanks for your followup! (Part 1)**
> > >
> > > Dear Reviewer Qf3U,
> > >
> > > So glad to see you for the follow-up! And thank you for agreeing with us that we should compare ImageNet results at a fair setup. We further respond to your concerns as follows.
> > >
> > > > `Q2`: "*From theoretical side, this work applied weight orthgonal regularizer to pruning..."*
> > >
> > > `A2`: As we responded previously, this evaluation may be a **misreading** of our paper. Our method is only *inspired by* orthogonality; yet literally, it is **not** a "*weight orthgonal regularizer*" -- see Eq. 3 and Fig. 2(c), there is *no* any orthogonality enforced on the parameters.
> > >
> > > We conceive your real concern might be: *If the method is not a "weight orthgonal regularizer", why does this paper mention orthogonal regularizers so much in this paper?* Here is a bit more background of how we developed the method -- This paper seeks better trainability for pruning. When we reviewed the literature about "trainability", we found many orthogonality regularizer papers; so naturally, there would be a naive idea of applying them to the pruning case, which leads to the 4 methods (L1 + KernOrth, L1 + OrthConv, KernOrth + L1, OrthConv + L1) in Tab. 1. This, we agree, would be incremental because it applies an *existing* idea to pruning. Rather, our idea (Fig. 2(c), Eq. 3) does *not* appear in prior works, so *ipso facto* it is *not* an incremental idea. Moreover, in terms of empirical performance, in the paper (Tab. 1), we have shown the really incremental ideas (L1 + KernOrth, L1 + OrthConv, KernOrth + L1, OrthConv + L1) do not work while our TPP works.
> > >
> > > ----
> > > `Q3/4`: "*From empirical side, the improvments of the proposed method on ImageNet-1k is not very significant. ... Yet, the improvements are still around 0.5%, hardly to say significant to me.*"
> > >
> > > `A3/4`: Regarding how to define "significant", clearly, it is up to different individuals. In our view, with all due responsibility, we are confident to say, in the area of filter pruning, 0.5% top-1 accuracy on ImageNet *is* a significant improvement, *especially when we compare to a very recent (CVPR'22) **SOTA** method*. (If we compare to a non-SOTA method, like Taylor-FO (CVPR'19) in Tab. 2, our method achieves 75.60% at 2.31x speedup, while Taylor-FO achieves 74.50% at only 1.82x speedup -- our method is **1.1% top-1 accuracy better at even greater speedup**).
> > >
> > > And notably, the speedup of the 0.5% improvement is 4x, which is not very large. As we've shown, the performance advantage of our method is *more obvious* in the *larger* sparsity regime in general. To show  this, like the CIFAR experiments (Tab. 1), we report pruning results of *more aggressive* pruning ratios with **ResNet50** on **ImageNet** (50% - 95%, *all layers* but the 1st Conv and last FC layer are pruned, including the downsample layers and the 3rd Conv in a residual block; we adopt uniform layerwise PRs below, which is not targeting performance but analysis):
> > >
> > > | PR (Pruning Ratio) | 0.5 | 0.7 | 0.9 | 0.95 |
> > > |---|---|---|---|---|
> > > | Sparsity/Speedup | 72.94%/3.63x | 89.34%/8.45x | 98.25%/25.34x | 99.34%/31.45x |
> > > | L1 | 71.25 | 66.02 | 47.96 | 33.21 |
> > > | TPP | 72.43 (+1.18) | 68.16 (+2.14) | 49.19 (+1.23) | 33.98 (+0.77) |
> > >
> > > As seen, at these greater speedups, **our method beats L1-norm pruning by 0.77%-2.14% top-1 accuracy**. Note, the fine-tuning stage uses *exactly the same* LR schedule (initial LR 0.01, 90 epochs), which is (among) the best fine-tuning LR schedule as far as we know, i.e., under a *competitive and fair* comparison setup. These results show our TPP indeed delivers **significant** improvements.
> > >
> > > ----
> > > `Q5 (1)`: "*The convergence curve seems interesting and thanks for the update! On CIFAR-10, when the pruning ratio is 30%~70%, the naive L1-norm pruning shows the same curve as the proposed method. This does not support the key argument of this work*"
> > >
> > > `A5 (1)`: The performance gap at sparsity 30% - 70% is not significant because at low sparsity regime, the trainability can be recovered by the network itself (DNNs are a complex system with *plasticity* [*1]), so different pruning methods (not just our method) under this setup do not present much difference -- to our knowledge, this is a widely-seen observation in the pruning literature.
> > >
> > > Such "low sparsity regime" depends upon the specific network/dataset (i.e., the tradeoff between network capacity and the task difficulty). E.g., on ResNet56+CIFAR10, PR 70% (Tab. 1), the speedup is 3.59x, while the performance gap is very marginal (TPP beats L1 by less than 0.1%). However, if you see the ImageNet case (Tab. 2), TPP can beat GReg-2 by 0.61% at 3.06x (they are comparable because of the same finetuning scheme), much more significant.
> > >
> > > In short, the small performance gap at 30% - 70% does *not* imply our method is not working or "*not support the key argument of this work*". It is a very common and normal phenomenon due to low sparsity.
> > >
> > > - [*1] 2018-WACV-Recovering from Random Pruning: On the Plasticity of Deep Convolutional Neural Networks

---

> > > ### Author Response · Authors · 2022-12-05
> > > **Sincerely expecting further discussions with Reviewer Qf3U**
> > >
> > > Dear reviewer Qf3U,
> > >
> > > After quite a long period, we finally finished all the experiments on ImageNet (thanks for your patience!). As you may notice, in the response `Thanks for your followup! (Part 1), A3/4` below, we show **our TPP beats L1 pruning by 0.77% - 2.14% top-1 accuracy on ImageNet** at larger speedups. We believe these gaps can probably be considered **significant improvements**.
> > >
> > > These results shows, one of the primary reasons that the advantage of our method appears "not significant" from your perspective in Tab. 2 is the *low sparsity* (we compare at these low sparsity regimes mainly because prior works had their experiments at this range), *not because our method does not work well*. At greater sparsities, the advantage of our method is much more pronounced.
> > >
> > > We are wondering if these new results can resolve your concern about the empirical effectiveness of our method. If so, would you generously consider improving the score as a fair recognition of our rebuttal efforts so far? Thank you so much!
> > >
> > > Sincerely,
> > >
> > > Authors

---

> > > > ### Author Response · Authors · 2022-12-08
> > > > **Expecting feedback**
> > > >
> > > > Dear reviewer Qf3U,
> > > >
> > > > Sorry for keeping reminding you (since the discussion deadline 12/12 is just around the corner). We are totally okay if your final decision is to keep the score. We are writing simply to know *what you make of the new ImageNet results and the background we posted* (see `A bit more background that may help you evaluate the performance of this paper` above) -- Do they make sense to you? And possibly, do they change your mind a bit?
> > > >
> > > > We value *the feedback much more* than a sole score. So we would really appreciate it if you could give us any feedback (like, if you think our point is not convincing, what kinds of experiments you think are missing to support the claim?). Your opinions are *rather important* for us to improve the work!
> > > >
> > > > Thank you!
> > > >
> > > > Sincerely,
> > > >
> > > > Authors

---

> > > > > ### Comment · Reviewer_Qf3U · 2022-12-08
> > > > > **I will raise my score**
> > > > >
> > > > > Thanks for the updated results! The new results look convincing. I will raise my score.

---

> > > > > > ### Author Response · Authors · 2022-12-08
> > > > > > **Thank you so much!**
> > > > > >
> > > > > > Dear reviewer Qf3U,
> > > > > >
> > > > > > *Thank you so much* for raising the score! We have learned a lot from your comments and are very glad that you agree our point is convincing. Thanks! Have a great day!
> > > > > >
> > > > > > Sincerely,
> > > > > >
> > > > > > Authors

---

> ### Author Response · Authors · 2022-11-15
> **Thank you! Responses to Reviewer Qf3U (Part 1)**
>
> Dear Reviewer Qf3U,
>
> ***Thank you so much*** for giving us so many detailed and very helpful comments. And thank you for the many listed strengths. We address your concerns as follows.
>
> > `Q1`: The novelty is incremental. Orthogonal regularization is not new in training deep network. Dating back to years ago, there were lots of discussions of weight regularization to make network more trainable, such as dynamic isometry constraint or nuclear norm regularizer. This work extends these ideas to network pruning, with some heuristic adaptations. For example, the correlation between selected important filters is not penalized in the proposed regularizer. BN norms of unimportant filters are considered. However, all these adaptations lack strong theoretical motivations. The authors did not provide any theoretical insight of why the proposed decorrelation plays a critical role in deep neural network pruning. From the numerical results, it seems that the improvement of the proposed method is also marginal on large-scale datasets.
>
> `A1`: We fully agree with Reviewer Qf3U on that "*orthogonal regularization is not new*", yet note, our regularization term is only *inspired by*, not equal to, orthogonality (in fact, our regularization term enforces *no* orthogonal regularization; see Fig. 2).
>
> We would also agree (to a certain degree) that our method is heuristic (by heuristic, we mean we do not present a rigorous math theorem to justify the design). Yet, **heuristic does not mean bad/lack of novelty**:
> - If we check out [the papers on neural network pruning](https://github.com/he-y/Awesome-Pruning), many of the most cited papers are heuristic, e.g., magnitude pruning [*1], deep compression [*2, *best paper*], L1-norm pruning [*3], SSL [*4], SFT [*5].
> - Magnitude pruning itself is actually the most heuristic pruning method, but this does not prevent it from being a *very strong* baseline method in the area (see https://arxiv.org/pdf/1902.09574.pdf).
> - The well-known lottery ticket hypothesis (LTH [*6, *best paper*]) is essentially based on magnitude pruning, which is heuristic -- So literally, **two ICLR best papers (ICLR'16 and ICLR'19) are based on a very simple heuristic pruning method**.
>
> Thus, we sincerely hope Reviewer Qf3U might not *cheapen* (in case this sounds like a strong word, we mean no offense) our paper simply because the method has heuristic parts.
>
> "*However, all these adaptations `lack strong theoretical motivations`. The authors did `not provide any theoretical insight` of why the proposed decorrelation plays a critical role in deep neural network pruning.*" -- We are sorry that we could not agree with this. The theoretical motivation is to preserve trainability originating from the kernel orthogonality. Fig. 2 has *clearly* shown how we arrive at our final design step by step. The key idea leap is from (b) to (c) in Fig. 2 -- this step we agree is a bit heuristic (again, heuristic != bad/lack of novelty), yet it intuitively makes sense, and importantly, the ablation study (Tab. 3) already justifies this design. The decorrelation scheme works well not because of some magic but simply because of *less regularization* (i.e., less constraint). Demanding orthogonality is unnecessary for the remaining weights; lifting such unnecessary constraints from the optimization problem and thus we see a good performance -- this is so straightforward to understand that we may not need to introduce any theorem.
>
> Lastly, we also pine for a method with "strong theoretical" basis as Reviewer Qf3U expects, yet Rome is not built in one day -- this would be a situation for any research area, especially for the new directions (as we state in the paper, this paper is the *first* trainability preserving method that can work on ImageNet). Thus, we humbly hope reviewer Qf3U may pay more attention to what we *have achieved* in this paper (good novelty as recognized by all other 3 reviewers + effective method + clear writing), instead of what we *have not* achieved (any paper would have sth that is to be done) -- note, **actually, all the other 3 reviewers agree that our paper *is novel***:
> - Reviewer p5ry: "this paper is of good novelty."
> - Reviewer aS72: "The method proposed in the paper is novel. The work is original."
> - Reviewer LVfu: "The paper makes novel and signi cant contribution", "The proposed method is theoretically sound", "The method is novel and solid"
>
> References:
> - [*1] 2015-NIPS-Learning both Weights and Connections for Efficient Neural Network
> - [*2] 2016-ICLR-Deep Compression: Compressing Deep Neural Networks with Pruning, Trained Quantization and Huffman Coding (**best paper**)
> - [*3] 2017-ICLR-Pruning Filters for Efficient ConvNets
> - [*4] 2016-NIPS-Learning Structured Sparsity in Deep Neural Networks
> - [*5] 2018-IJCAI-Soft Filter Pruning for Accelerating Deep Convolutional Neural Networks
> - [*6] 2019-ICLR-The Lottery Ticket Hypothesis: Finding Sparse, Trainable Neural Networks (**best paper**)

---

### Official Review · Reviewer_aS72 · 2022-10-25

**Confidence:** 3
**Correctness:** 2
**Technical Novelty And Significance:** 2
**Empirical Novelty And Significance:** 2
**Recommendation:** 6

**Clarity, Quality, Novelty And Reproducibility:**

The logic of the paper is easy to follow, but some motivations are not clear.
The method proposed in the paper is novel.
The work is original.

**Strength And Weaknesses:**

Strength:
1. The trainability of pruned models is interesting, and may provide a new perspective to pruning, and even to the whole deep learning field.
2. The proposed pruning method is easy to implement, and produces good experimental results.

Weakness:
1. The relationship between trainability and dynamical isometry is not clear. I find no evidence to indicate this point, i.e. how can trainability be expressed by the orthogonal property of channels?
2. The necessity of contraining BN parameters is not shown. The reason in the paper is that "Although unimportant weights are enforced with regularization for sparsity, their magnitude can barely be exact zero, making the subsequent removal of filters biased". However, if the pruned weights are not zeros, how to achieve the speedup of models during  the inference phase?

**Summary Of The Paper:**

The authors present trainability prerseving pruning (TPP), considering to maintain trainability for the pruned networks. The authors construct two regularization terms to achieve this goal. TTP decorrelates the pruned weights from the kept weights, thus achieves non-trivial improvement in pruning process.

**Summary Of The Review:**

The paper focuses on an interesting topic, and proposes a working solution. However, the motivation of the methods and some specific techniques are not very clear. Therefore, I think this paper should be weakly rejected.

---

> ### Author Response · Authors · 2022-11-14
> **Thank you! Responses to Reviewer aS72**
>
> Dear Reviewer aS72,
>
> ***Thank you so much*** for helping us improve the paper! We are so glad and grateful that you agree with us that (1) the trainability issue of pruning is "*interesting and may provide a new perspective to pruning, and even to the whole deep learning field*" (2) Our method is "*easy to implement*", and delivers "*good experimental results*".
>
> We address your concerns point by point as follows.
>
> > `Q1`: The relationship between trainability and dynamical isometry is not clear. I find no evidence to indicate this point, i.e. how can trainability be expressed by the orthogonal property of channels?
>
> `A1`: We add a few backgrounds as the common ground.
>
> Trainability, by its name, means the *ability to train a neural network*. It is usually studied in the area of deep learning theory, e.g., in the abstract of "Disentangling Trainability and Generalization in Deep Neural Networks" (ICML'20), the authors state: "*A longstanding goal in the `theory of deep learning` is to characterize the conditions under which a given neural network architecture will be `trainable`, and if so, how well it might generalize to unseen data*".
>
> Given a predefined network architecture, *if the initialization scheme is proper*, the loss function would be constantly decreasing (at least in the early stage) -- then this network is deemed "trainable". So as you may notice, **trainability is very closely related to initialization** (many of us may overlook the central role of initialization in deep learning applications today because the current DL platforms have implemented very mature interfaces for us. E.g., for Pytorch, when we define a layer using `torch.nn.Conv2d`, underneath, it has been initialized by the [Kaiming uniform initialization](https://github.com/pytorch/pytorch/blob/06486cd0087200e08ebb8a9518e064251c7c5309/torch/nn/modules/conv.py#L150) -- this step looks simple today, but historically, the initialization problem has baffled researchers for *more than a decade*, to say the very least; see [*1-*5] ).
>
> Then, speaking of initialization, a consensus is that, a good initialization should be *norm-preserving* [*6, (Saxe et al., 2014)] among layers, especially for the gradients -- this is easy to see, as norm-preservation implies the gradients will not be magnified or attenuated much, i.e., we would *not* see gradient exploding or vanishing. For linear FC layers, the weights are simply a 2d matrix. It is easy to see with Linear Algebra that the *orthogonal matrix* meets such "norm-preserving" property (i.e., the Eq. 1 in our paper) -- this is how trainability is connected with orthogonality. For Conv layers, we typically reshape the 4d weights into 2d matrix (following previous kernel orthogonality papers -- (Xie et al., 2017; Huang et al., 2018; 2020)), thus, trainability is now connected to orthogonality of the filters/channels.
>
> Hope our explanation above can resolve your concern. *Thanks a lot* for letting us know that we actually did not explain the connection very clearly! We shall add the above discussions to the revised version.
>
> - [*1] 2010, AISTATS, Understanding the difficulty of training deep feedforward neural networks
> - [*2] 2013, ICML, On the importance of initialization and momentum in deep learning
> - [*3] 2016, ICLR, All you need is a good init
> - [*4] 2015, Data-dependent initializations of convolutional neural networks
> - [*5] 2015, CVPR, Delving deep into rectifiers: Surpassing human-level performance on imagenet classification
> - [*6] 2021, TPAMI, Norm-Preservation: Why Residual Networks Can Become Extremely Deep?
>
> > `Q2`: The necessity of contraining BN parameters is not shown. The reason in the paper is that "Although unimportant weights are enforced with regularization for sparsity, their magnitude can barely be exact zero, making the subsequent removal of filters biased". However, if the pruned weights are not zeros, how to achieve the speedup of models during the inference phase?
>
> `A2`: "*if the pruned weights are not zeros, how to achieve the speedup of models during the inference phase?*" -- There might be a misunderstanding here. The whole algorithm pipeline is: Step 1. apply the proposed regularization term to a pretrained model (pretrained model is typically available already, like the torchvision models we used); Step 2. enforce the pruning action -- this is the instant weight removal (or equivalently, zeroing out the pruned weights); Step 3. fine-tune the output model in Step 2.
>
> When we say "their magnitude can barely be exact zero", we mean *the regularization process, i.e., Step 1*. In Step 2, the pruned weights will be made exact zeros, thus we can achieve speedup during inference.
>
> **Again, thank you so much for the very helpful reviews! Let us know if you have *any* further questions. We are *actively available* until the end of this rebuttal period. Looking forward to hearing back from you!**

---

> ### Author Response · Authors · 2022-11-17
> **Sincerely expecting further discussions with Reviewer aS72**
>
> Dear Reviewer aS72,
>
> We greatly thank you for helping us improve our paper so far! Given the ICLR discussion deadline that allows us to update the submission pdf (11/18) is approaching, we genuinely hope to have a further discussion with you to see if our responses resolve your concerns, esp. if you need us to add more results to the submission.
>
> To address your major concern ("*but some motivations are not clear*"), we now have added further explanations to clarify the connection between trainability and the orthogonality of filters.
>
> Should the motivation issue have been clarified, meanwhile, many aspects of the paper are valuable according to your comments (*"The logic of the paper is **easy to follow**. The method proposed in the paper is **novel**. The work is **original**. The paper focuses on an **interesting** topic, and proposes a **working** solution"*), we were wondering, if you could generously consider raising the score even a little bit as a fair recognition of our rebuttal efforts so far? *Thank you so much!*
>
> Sincerely,
>
> Authors

---

> ### Author Response · Authors · 2022-12-04
> **Sincerely expecting further discussions with Reviewer aS72 (2nd call)**
>
> Dear Reviewer aS72,
>
> We greatly thank you for helping us improve our paper so far! Given the ICLR discussion deadline (12/12) is approaching, **we genuinely hope to have a further discussion with you** to see if our responses resolve your concerns. Your opinion is *rather important* to us to get the paper better. Thank you!
>
> Sincerely,
>
> Authors

---

> > ### Comment · Reviewer_aS72 · 2022-12-07
> > **Reply for author's responses**
> >
> > Thank you for your feedback. I still have one question.
> >
> > In the reply for Question 1, I agree that orthogonal matrices can bring the norm preserving property. In my opinion, the orthogonal property is only one of the characteristics of orthogonal matrices. They also have to keep the property of eigenvalues. So if the weights only orthogonal property, are they trainable?

---

> > > ### Author Response · Authors · 2022-12-07
> > > **Thank you for following up!**
> > >
> > > Dear Reviewer aS72,
> > >
> > > Thank you so much for following up and raising the score!
> > >
> > > "*In my opinion, the orthogonal property is only one of the characteristics of orthogonal matrices. They also have to keep the property of eigenvalues.*" -- This understanding is correct. Per the definition of [orthogonal matrix](https://www.wikiwand.com/en/Orthogonal_matrix), in addition to orthogonality (i.e., the inner product of row/column vectors is zero), their L2-norm has to be 1 (i.e., orthonormal).
> > >
> > > "*So if the weights only orthogonal property, are they trainable?*" -- The weights can be forced to have the orthogonal property **only at the initialization**. They are trainable. When the training starts, the orthogonal property cannot be maintained anymore. So previous works (Xie et al., 2017; Huang et al., 2018; 2020) actually try to integrate a regularization term to encourage the orthogonal property over the whole training instead of just focusing on the initialization.
> > >
> > > Let us know if you have more questions! Thanks!
> > >
> > > Sincerely,
> > >
> > > Authors

---

### Official Review · Reviewer_p5ry · 2022-10-26

**Confidence:** 4
**Clarity, Quality, Novelty And Reproducibility:** this paper is well written and of goo…
**Correctness:** 4
**Technical Novelty And Significance:** 3
**Empirical Novelty And Significance:** 3
**Recommendation:** 6

**Strength And Weaknesses:**

Pros
This paper proposes a post process for pertained network to get  more trainable sparse weights for finetuning. Experiments verifies the trainability of the sparse weight via comparisons with different finetuning  learning rate.  Extensive experiments on several datasets illustrates the efficacy of the proposed method. In addition, the authors also consider the effect of batch normalization parameters and adds to the regularization term.

Cons
The authors illustrate the ablation with regularizing  BN weight and without regularizing BN weight. However, I do not find the results of using the regularization term of BN parameters alone. Could the authors show this result?


Question
As shown in the paper, the proposed TPP method is a post-process algorithm for a pertained model. After the TPP process, the sparse weight is finetuned to get a sparse model with decent performance. I am wondering whether the TPP algorithm can be combined with the pretraining? In addition,  the TPP post process also takes the classification loss into consideration, so it can be considered as the part of finetuning. It would be better to have the experiments with similar finetuning cost to compare TPP with other methods in a more fair way. Whether the proposed TPP can be useful for finding lottery ticket subnetwork in a filter level?

**Summary Of The Paper:**

This paper focuses on preserving the trainability of network after pruning. As proposed by the authors, the trainability of pruned networks can be affected with the phenomenon of being less robust to finetuning learning rate. To alleviate this issue, the authors proposes the trainability preserving pruning (TPP) that can maintain the trainability during pruning. The authors suggest that the dependency between weights may cause the issue. So in this paper, the authors propose to decorrelate the pruned weights and the kept weights. In detail, the authors propose to regularize the gram matrix of weights in the way that the correlation of between the kept and pruned weights approaches 0. In addition, the authors suggest that the parameters of BatchNormalization should be explicitly considered for pruning by regularizing the parameters. Experiments on MNIST, Cifar and ImageNet verifies the efficacy of the proposed methods. On ImageNet classification task  with ResNet50, the proposed TPP  illustrates significant improvement compared with other SOTA methods.

**Summary Of The Review:**

please refer the comments listed above.

---

> ### Author Response · Authors · 2022-11-15
> **Thank you! Responses to Reviewer p5ry (Part 2)**
>
> > `Q3`: In addition, the TPP post process also takes the classification loss into consideration, so it can be considered as the part of finetuning. It would be better to have the experiments with similar finetuning cost to compare TPP with other methods in a more fair way.
>
> `A3`: We understand and also agree with this potential concern. Here we address it: First, note that the results of "L1 + KernOrth", "L1 + OrthConv", "KernOrth + L1", "OrthConv + L1" in Tab. 1 also involves training (the KernOrth/OrthConv is essentially a regularization training), which takes *50k* iterations.
> - Our TPP takes 100k iterations based on the default hyper-parameter setup (see Appendix Tab. 6), so to make a fair comparison, we decrease the *regularization update interval $K_u$* to 5, making the regularization of TPP also take 50k iterations.
> - Meanwhile, we add 128 finetuning epochs (50k / 391 iters per epoch ≈ 128 epochs) to the "L1" and "L1 + OrthP" methods (when their finetuning epochs are increased, the LR decay epochs are proportionally scaled); plus the original 120 epochs, the total epochs are 248 now.
>
> Now all the comparison methods in Tab. 1 have the same training cost (i.e., the same total epochs). The new results of `L1`, `L1+OrthP`, `TPP` under this strict comparison setup are presented below:
>
> Res56+CIFAR10, finetuning init LR 0.01, **all methods keep ~248 total training epochs**:
> | Pruning ratio (PR)  | 0.3 | 0.5 | 0.7 | 0.9 | 0.95 |
> |---|---|---|---|---|---|
> | L1 |  93.65$_{\pm0.14}$ |   93.38$_{\pm0.16}$ | 92.11$_{\pm0.16}$ | 87.17$_{\pm0.26}$ |  83.94 $_{\pm0.45}$ |
> | L1+OrthP | 93.58$_{\pm0.03}$ | 93.30$_{\pm0.10}$ | 91.69$_{\pm0.13}$ | 85.75$_{\pm0.26}$ | 82.30$_{\pm0.20}$ |
> | TPP | 93.76$_{\pm0.10}$ | 93.45$_{\pm0.05}$ | 92.42$_{\pm0.14}$ | 89.54$_{\pm0.08}$ | 85.98$_{\pm0.29}$ |
>
> **Observations**:
> - For L1 pruning, more finetuning epochs do not always help. Comparing these results to Tab. 1 in the paper, we may notice at small PR (0.3, 0.5), the accuracy drops a little (this probably is due to overfitting -- when the PR is small, the pruned model does not need so many epochs to recover; while too long training triggers overfitting). For larger PR (like 0.95), more epochs help quite significantly (improving the accuracy by 0.91%).
> - L1 + OrthP still underperforms L1, same as in Tab. 1.
> - Despiting using fewer epochs, TPP is still pretty robust -- Compared to Tab. 1 in the paper, the performance varies by a *very marginal* gap (~0.1%, within the std range, so not a statistically significant gap). In general, TPP is still *the best* among all the compared methods, and, the advantage is more obvious at larger PRs, implying TPP is more valuable at the aggressive pruning regime.
>
> Thanks for letting us know about this potential fairness issue. We would include these results in the revised version to clarify this.
>
> > `Q4`: Whether the proposed TPP can be useful for finding lottery ticket subnetwork in a filter level?
>
> `A4`: Thanks for pointing out this very interesting direction! At this point, we are not very sure about this (esp. when we do not have any empirical studies as support), so we refrain from making bold claims. The primary obstacle we think is: To our best knowledge, filter-level winning tickets (WT) are still hard to find even using the original LTH pipeline. Few attempts in this direction have succeeded -- E.g., this paper (https://arxiv.org/pdf/2202.04736.pdf, ICML'22) tried, but they can only achieve a bit marginal sparsity (~30%) with filter-level WT (see their Fig. 3, ResNet50 on ImageNet) while weight-level WT typically can be found at over 90% sparsity. This said, we do think this paper can contribute in that direction since preserving trainability is also a central issue in LTH too.
>
> We would add this valuable discussion in the revised version.
>
> **Again, thank you so much for the detailed and very constructive comments! Let us know if you have *any* further questions. We are *actively available* until the end of this rebuttal period.**

---

> ### Author Response · Authors · 2022-11-15
> **Thank you! Responses to Reviewer p5ry (Part 1)**
>
> Dear Reviewer p5ry,
>
> ***Thank you so much*** for the very constructive comments and the positive score!
>
> We address your concerns and questions as follows. *Like the experiments reported in the paper, we promise to release all the experiment logs/checkpoints during this rebuttal*.
>
> > `Q1`: The authors illustrate the ablation with regularizing BN weight and without regularizing BN weight. However, I do not find the results of using the regularization term of BN parameters alone. Could the authors show this result?
>
> `A1`: The results of *only* applying the BN regularization for Tab. 4 in the paper are presented below. As usual, each result is averaged by 3 random runs.
>
> Res56+CIFAR10, finetuning init LR 0.01, before finetuning:
> |  | 0.3 | 0.5 | 0.7 | 0.9 | 0.95 |
> |---|---|---|---|---|---|
> | TPP (only BN reg) | 92.40$_{\pm0.30}$ | 91.95$_{\pm0.04}$ | 90.26$_{\pm0.23}$ | 26.79$_{\pm2.19}$ | 10.50$_{\pm0.63}$ |
>
> **Observations**: If we compare this table to Tab. 4 in the paper, we will notice the results here are the worst. This is as expected since only using BN reg means the majority of the learnable params (weights + biases) are not attended. They answer for the majority of trainability, so not regularizing them leads to the worst performance here -- This table actually can be taken as another sanity-check to justify our motivation/method.
>
> > `Q2`: As shown in the paper, the proposed TPP method is a post-process algorithm for a pertained model. After the TPP process, the sparse weight is finetuned to get a sparse model with decent performance. I am wondering whether the TPP algorithm can be combined with the pertaining?
>
> `A2`: TPP definitely can be applied to the pretraining stage, namely, pruning a *random* model (thus, it would fall into the *sparse training* category instead of the conventional pruning after training). We provide the results below, along with the results of L1-norm pruning applied to a random model as comparison.
>
> Res56+CIFAR10, finetuning init LR 0.01:
> |  | 0.3 | 0.5 | 0.7 | 0.9 | 0.95 |
> |---|---|---|---|---|---|
> | L1 | 88.98$_{\pm0.13}$ | 88.79$_{\pm0.18}$ | 87.60$_{\pm0.21}$ | 85.09$_{\pm0.09}$ | 82.68$_{\pm0.32}$ |
> | TPP | 91.93$_{\pm0.13}$ | 91.27$_{\pm0.16}$ | 90.36$_{\pm0.16}$ | 87.60$_{\pm0.09}$ | 85.36$_{\pm0.18}$ |
>
> **Observations**:
> - First, applying TPP to a random model *underperforms* those reported in the paper (applying TPP to a *pretrained* model). This makes sense since the pretrained model is well-known to provide a good initialization for pruning (and this is simply why pruning is typically conducted onto a pretrained model).
> - TPP still surpasses L1-norm pruning, suggesting the effectiveness of TPP over other pruning methods can translate to the case of sparse training.
>
> We shall add these results to the revised version and discuss how TPP can be used in the case of sparse training. Thanks for giving this wonderful suggestion!

---

> ### Author Response · Authors · 2022-12-06
> **Expecting more feedback from reviewer p5ry**
>
> Dear reviewer p5ry,
>
> Thank you so much for being with us so far! We genuinely hope to hear more feedback from you -- Have our responses resolved your concerns? Would you consider raising the score even a little bit as a fair recognition of our rebuttal efforts so far (considering based on our responses, we seem to have addressed your concerns)? Thank you!
>
> Sincerely,
>
> Authors

---

> > ### Comment · Reviewer_p5ry · 2022-12-11
> > **Thanks!**
> >
> > Thanks! the rebuttal addressed my concerns.  (sorry for the late reply).
> >
> > best,

---

### Author Response · Authors · 2022-11-16
**To All Reviewers: thank you and hope you may take a quick look at our responses**

Dear Reviewers,

*Thank you so much* for spending your precious time reviewing our paper. We have luckily got very informative feedback from your comments so far, which are indispensable for us to polish the paper to a better version.

Meanwhile, as you may notice, we have responded to all of you (although some results are to be updated shortly). We are now writing simply to wonder **if you could spend a few minutes taking a quick look at our responses in case we misread your comments or conducted the wrong experiments**. Also, based on our responses, **if you think there should be more experiments/analyses, please let us know *asap* so that we can have enough time to finish them early**.

We would also really appreciate it if you could give us a little more feedback based on our responses (e.g., *do our responses resolve your concerns?*)

Again, thank you all for being with us so far to improve the paper!

Sincerely,

Authors,11/16

---
 `11/19 Update`: Dear reviewers, thanks a lot for helping us with the paper! As you may notice, we have updated the submission pdf. New results (mainly figures, as it is not easy to present figures directly on OpenReview) are included in the Appendix (highlighted in magenta color). We are looking forward to your further feedback! -Best, Authors

---

### Author Response · Authors · 2022-12-06
**A bit more background that may help you evaluate the performance of this paper**

We summarize a bit the reviewing status quo after the discussions with reviewers, for the convenience of ACs and any readers who are interested in this paper.

As of 12/6, we got 6/6/6/3. One reviewer (aS72) raised the score from 5 to 6; other reviewers did not change. Reviewer Qf3U is the most negative one. He/she rated 3 based on two reasons: limited novelty and not significant performance. As we've explained, the "limited novelty" evaluation of our method (Qf3U says our method is a  "*orthogonal regularizer*", a natural extension from existing works) is probably a *misinterpretation*. Our method demands *no orthogonality* on the kept parameters, which literally is **not** an "*orthogonal regularizer*". Besides, all the other three reviewers agree that our paper *is novel*.

Then, the major concern is only about the empirical performance, which has two parts:
- (1) At even greater speedups (like above 3x), our new results with ResNet50 on ImageNet have shown that our method can beat L1-norm pruning by 0.77% to 2.14% accuracy, which is significant. But reviewer Qf3U is concerned that the performance against the unpruned model drops too much, making the method not very useful in these cases.
- (2) At small speedups (like 2x-3x), our method does not show much advantage over other methods (top-1 accuracy improvement on ImageNet is <= 0.5%).

Reviewer LVfu also has this concern.

First, we *agree* on these observations. What we may disagree on is, how should we interpret them *fairly* when evaluating our paper?

- About (1), our point is, **the practical value of a pruning method is always in a *relative* sense, relative to the other methods instead of the unpruned model** (against the unpruned model, at large enough sparsity, the performance would always drop much). E.g., if we target a model with 60% top-1 accuracy by pruning ResNet50 (torchvision accuracy 76.15%) on ImageNet; our method achieves 65%, while others achieve 60%. Although our method drops by 11.15% accuracy, which is definitely a significant drop, yet *as long as our method is better than other methods, it is still useful*. Thus, we would say, even if our method drops much at these greater speedups, it is still useful since other methods drop *even more* than we do.

- About (2), this is actually the case where we would like to provide more background. As far as we know, different filter pruning methods actually perform *pretty closely* at low speedups, *if they are benchmarked at a **fair** comparison setup*.

Below is a showcase of L1-norm pruning (probably the most simple filter pruning method) compared to other recent top-performing methods. The L1-norm pruning is fine-tuned with a better LR schedule (the original finetuning LR schedule in the L1 pruning paper -- fixed LR 0.001, 20 epochs -- is severely suboptimal; we use 90-epoch finetuning, initial LR 0.01, decayed at 30/60/75 epoch. This is also the schedule we use to report the new results of ResNet50 on ImageNet during this rebuttal):

| Method | Pruned accuracy (%) | Speedup |
|---|---|---|
| SFP (IJCAI 2018) | 74.61 | 1.72x |
| DCP (NeurIPS 2018) | 74.95 | 2.25× |
| GAL-0.5 (CVPR 2019) | 71.95 | 1.76× |
| Taylor-FO (CVPR 2019) | 74.50 | 1.82× |
| CCP-AC (ICML 2019) | 75.32 | 2.18× |
| ProvableFP (ICLR 2020) | 75.21 | 1.43× |
| HRank (CVPR 2020) | 74.98 | 1.78x |
| GReg-1 (ICLR 2021) | 75.16 | 2.31× |
| GReg-2 (ICLR 2021) | 75.36 | 2.31× |
| CC (CVPR 2021) | **75.59** | 2.12x |
| L1-norm (ICLR 2017) (better finetuning LR) | 75.24 | 2.31× |
|
| Factorized (CVPR 2019) | 74.55 | 2.33× |
| LFPC (CVPR 2020) | 74.46 | 2.55× |
| GReg-1 (ICLR 2021) | 74.85 | 2.56× |
| GReg-2 (ICLR 2021) | **74.93** | 2.56× |
| CC (CVPR 2021) | 74.54 | 2.68× |
| L1-norm (ICLR 2017) (better finetuning LR) | 74.77 | 2.56× |

As you may notice, at around 2x ~ 3x speedups, the gap between the best method and L1 pruning is **only 0.16% - 0.35%**. Namely, **(nearly) all the methods have the same "*not significant improvement*" problem, not just our method**. Someone may see a paper claiming a significant improvement (e.g., >0.5%) against L1-norm pruning at ~2x speedup with ResNet50 on ImageNet. To our knowledge, the performance improvement is mostly due to a better finetuning process (e.g., longer epochs, better LR schedules, and more advanced data augmentations) instead of the pruning algorithm itself.

If this is a "problem" (we are not even sure if this should be regarded as a *problem*, because probably it is just what it is for all pruning methods, a natural phenomenon) for (nearly) all methods, we don't think it should be taken as a disadvantage *singled out for our paper* for fairness. Of course, this is the way we interpret the results. We know different researchers have their own standards. We just want the ACs and other readers to be aware of the benchmarking situation in the current network pruning area and make fair evaluation.

Any comments about the above discussion are welcome!

Thanks,

Authors

---

> ### Author Response · Authors · 2022-12-08
> **Reviewing status summary update (as of 12/08)**
>
> `12/08 Update`: The last negative reviewer Qf3U agrees that our new results and statements are convincing, the score thus increased from 3 to 6 (so now the scores are 6/6/6/6). Thanks!
>
> So far, we've got so much very informative feedback from all reviewers. Great thanks to all of them (and the ACs for arranging the reviewing process)!
>
> Sincerely,
>
> Authors

---

### Decision · Program_Chairs · 2023-01-20

**Decision:**

Accept: poster

**Justification For Why Not Higher Score:**

The proposed method is not backed up by theory.

**Justification For Why Not Lower Score:**

N/A

**Metareview: Summary, Strengths And Weaknesses:**

The paper proposes the trainability preserving pruning (TPP) that can maintain the trainability during pruning. This is achieved by regularizing the gram matrix of convolutional filters to decorrelate the pruned filters from the retained filters. In addition to the convolutional layers, the batch normalization parameters are also regularized to preserve the trainability of the network. There was an extensive discussion over the paper, and I communicated with the reviewers multiple times during the discussion period. The authors eagerly addressed most of the concerns about novelty and provided new large-scale experimental results on ImageNet which further confirmed the effectiveness of the proposed method. Hence, I recommend acceptance.

**Note From Pc:**

if the above contains the word "oral" or "spotlight" please see: "oral" presentation means -> notable-top-5% and "spotlight" means -> notable-top-25%. As stated in our emails, we are disassociating presentation type from AC recommendations